# Awakening Collective Wisdom: Elevating Super-Resolution Network Generalization through Cooperative Game Theory

## Abstract

Improving the generalization capability of image super-resolution algorithms is a fundamental challenge when deploying them in real-world scenarios. Prior methods often relied on the assumption that training on diverse data can improve generalization capabilities, leading to the development of complex degradation models that simulate real-world degradation. Unlike previous works, we present a novel training strategy grounded in cooperative game theory to improve the generalization capacity of existing image super-resolution algorithms. Within this framework, we conceptualize all neurons in the network as participants engaged in a cooperative relationship, where their collective responses determine the final prediction. As a solution, we propose to awaken suppressed neurons that hinder the generalization capability through our **E**rase-and-**A**waken **T**raining **S**trategy (**EATS**), thus fostering equitable contributions among all neurons and effectively improving generalization performance. EATS offers several compelling benefits. *1) Seamless integration with existing architectures*: It integrates with existing networks to enhance their generalization capability for unseen scenarios. *2) Theoretically feasible strategy*: We theoretically prove the effectiveness of our strategy in enhancing the Shapley value (reflecting each participant's contributions to prediction). *3) Consistent performance improvements*: Comprehensive experiments on various challenging datasets consistently demonstrate performance improvements when employing our strategy. The code will be publicly available.

## 1 Introduction

Single image super-resolution (SR) is a classical low-level vision task that focuses on restoring a high-resolution (HR) image from a low-resolution (LR) version. Recently, deep learning-based SR (Dong et al., 2015; Zhang et al., 2018b; 2019; 2018a; Chen et al., 2022; Zhang et al., 2022; Wang et al., 2023; Zhang et al., 2022) algorithms have made significant efforts in synthetic environments. However, overfitting to specific degradations in synthetic scenarios leads to poor generalization performance when deploying these algorithms in real-world scenarios due to domain gaps.

In response to the mentioned challenge, some efforts have been devoted to diversify the synthetic training data to encompass the broader space of real-world degradation, thereby enhancing the generalization capabilities of SR models. For example, BSRGAN (Zhang et al., 2021) introduced a complex degradation model involving a random shuffle of degradation orders, while Real-ESRGAN (Wang et al., 2021) introduced second-order degradation techniques. However, these efforts operate under the assumption that diversifying the training data can indeed improve generalization capabilities. In contrast to previous works, we contemplate a shift in focus from data to optimization, aiming to present a flexible training paradigm that fosters the generalization capability of algorithms.

**Observation.** To achieve this goal, we conduct an initial exploration into the properties of algorithms when applied to in-the-wild scenes. The co-adaption phenomenon, discussed in this work, Hinton et al. (2012), provides fundamental insights for our investigation. It suggests that training over-parameterized networks often leads to the over-activation of specific neuronal connections while inhibiting others. However, when exposed to unseen real-world scenarios, these established

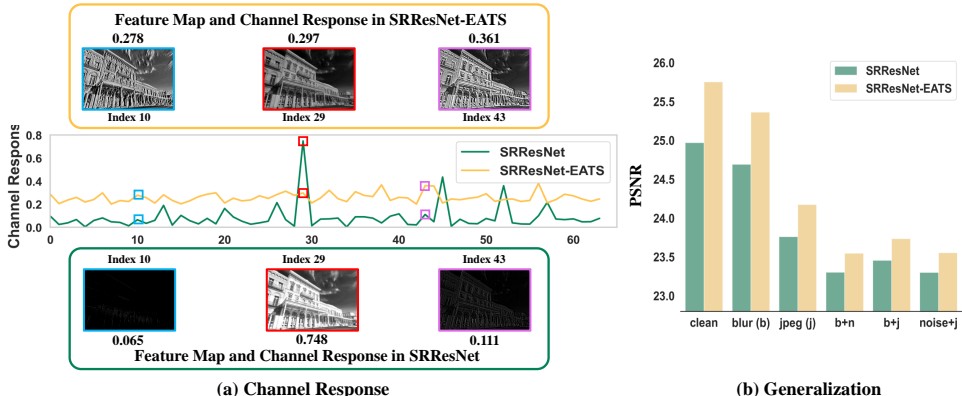

Figure 1: The Erase-and-Awaken Training Strategy (EATS) mitigates the co-adaption problem by promoting equitable contributions among all filters, thereby improving generalization performance. (a) Co-adaption in SR: We analyze channel responses in unseen scenarios, averaging channel responses from the $2^{nd}$ block in SRResNet (Ledig et al., 2017) (trained on Five5K (Bychkovsky et al., 2011)) across 100 randomly sampled unseen images. We visualize the feature maps and responses of channel index 10, 29, and 43. It reveals the co-adaption phenomenon (Hinton et al., 2012), where a few channels are highly activated while others are inhibited, resulting in poor generalization. In contrast, integrating EATS with SRResNet achieves equitable and activated channel responses. (b) Generalization performance: Results on Set5 with ×4 scaling under different degradation settings, showcasing the effectiveness of EATS in improving the network's generalization capacity.

connections perform inadequately than on the training data, thus limiting the network's generalization capacity. To verify this phenomenon within the SR task, we investigate the average response of each channel within the same layer on unseen data scenarios. As depicted in Fig. 1(a), we observe that only a few channel responses are highly activated while the majority remain inhibited. These dormant channels represent a critical factor hindering algorithms' generalization performance. This finding leads to a pivotal question: "How can we 'awaken' these dormant channels and encourage equitable contributions among all filters to prediction for improving the generalization capacity?"

**Solution.** In this paper, we introduce cooperative game theory to tackle the generalization challenges in SR algorithms. We present a novel training strategy, denoted as Erase-and-Awaken Training Strategy (EATS), which offers a distinctive perspective by viewing all neurons within the network as active participants engaged in a collaborative relationship. Within these networks, intertwined with complex neuronal connections, their combined responses harmoniously converge to ascertain the ultimate prediction. To awaken the inhibited participants that hinder generalization performance, we propose to randomly perturb the responses of the neurons and maximize their contributions to the prediction. As presented in Fig. 1(a), averaged responses of the initially inhibited channel (index 10 and 43) have been awakened while an abnormally highly activated channel (index 29) was suppressed. It demonstrates the ability of EATS to promote equitable contributions from all channels to the network's prediction but also attain consistent channel response distribution on unseen images. Furthermore, in Fig. 1(b), we provide quantitative evidence to show the effectiveness of our EATS strategy in boosting algorithm's generalization performance on unseen multi-degraded scenarios.

The main contributions of this work are summarized as follows: (1) We propose a novel training strategy grounded in cooperative game theory, fostering equitable contributions among all neurons to enhance the generalization capacity of SR algorithms. (2) Our EATS can seamlessly integrate into existing super-resolution networks, providing a flexible and effective training paradigm. Theoretical proof substantiates the effectiveness of our strategy in improving the Shapley value (measuring each participant's contribution to predictions). (3) Extensive experiments conducted on various challenging datasets consistently demonstrate performance improvements via employing our strategy.

## 2 RELATED WORK

**Super-Resolution.** Image super-resolution task focuses on restoring a high-resolution image from its corrupted low-resolution counterpart. Conventional SR networks (Dong et al., 2014; Dai et al.,

2019a; Kim et al., 2016; Lim et al., 2017; Wang et al., 2018a; Yuan et al., 2018; Dai et al., 2019b; Xia et al., 2022; Lai et al., 2017; Sajjadi et al., 2017; Johnson et al., 2016; Ma et al., 2020; Liu et al., 2020; Zhou et al., 2020; Magid et al., 2021) are typically trained on image pairs generated via bicubic interpolation. However, it may induce overfitting towards the specific degradation, resulting in limited generalization in real-world degradations. As a solution, a blind SR paradigm involving degradation prediction and super-resolution is introduced to tackle complex real-world scenarios. IKC Gu et al. (2019) introduces a predict-and-correct principle, *i.e.*, iteratively correcting the estimated kernel based on the previous SR results. DAN (Huang et al., 2020) proposes an alternating optimization algorithm for estimating blur kernel and restoring super-resolved image iteratively. Nevertheless, these techniques are confined to addressing conditions outlined within the predefined degradation model and often exhibit sub-optimal generalization when encountering images that deviate from the degradation model. Therefore, the alternative approach involves enriching the training set with random combinations of diverse degradations, such as the high-order degradation process in Real-ESRGAN (Wang et al., 2021) and the degradation shuffle strategy in BSRGAN (Zhang et al., 2021). However, the aforementioned methods exclusively simulate real-world scenarios by predicting degradations within the predefined model and augmenting the diversity of training data while neglecting the exploration of more streamlined training strategies.

**Generalization in low-level vision.** Since the degradations encountered during the training phase cannot faithfully simulate the complicated degradations inherent in real-world scenarios, the challenge of generalization in low-level tasks has garnered significant attention. In terms of the generalization assessment, Liu et al. (2021) introduces a deep degradation representation as an approximate evaluation metric for measuring generalization ability, where the worse the degradation clustering effect means the better generalizability. SRGA Liu et al. (2022) delves into the statistical attributes of internal features within deep networks to measure their generalization capability. Moreover, many researchers are dedicated to improving the generalizability of networks. Kong et al. (2022) broke the common sense that dropout cannot be effectively applied to low-level vision and explored its working mechanism in SR. Li et al. (2023) introduced a causality training strategy that focuses on learning the distortion-invariant representations, thereby enhancing the generalization capacity.

## 3 METHOD

### 3.1 PRELIMINARY

Shapley value serves as a fundamental tool within cooperative game theory for credit allocation. In cooperative game theory, a game is defined by a set function where each subset's value represents the profit when the associated players participate. Given a cooperative game model, the game consists of a set $\mathcal{N}$ with $|\mathcal{N}|$ individual players, denoted as $\mathcal{N} = \{n_i\}_{i=1}^{i=|\mathcal{N}|}$. The game's profit is assessed using a specific metric denoted as $v$, which takes subsets $\mathcal{M} \subseteq \mathcal{N}$ as input and generates a profit score. Formally, Shapley values for the $i^{th}$ player are calculated as

$$\phi_{n_i}(\mathcal{M}, v) = \sum_{\mathcal{M} \subseteq \mathcal{N} \setminus \{n_i\}} \frac{|\mathcal{M}|!(|\mathcal{N}| - |\mathcal{M}| - 1)!}{\mathcal{N}!} [v(\mathcal{M} \cup \{n_i\}) - v(\mathcal{M})]. \tag{1}$$

Intuitively, Eq. 1 quantifies the contribution of a specific player $n_i$, to the overall profits when introducing this player. This quantification averages across all conceivable subsets where $n_i$ can be included. The Shapley value adheres to several rational properties: (1) Additivity: The Shapley value of each player sum to the profits when all players participate. (2) Symmetry: Players with equivalent contributions receive equal Shapley value. (3) Dummy players: Players who make no contribution receive a Shapley value of zero. These properties ensure that Shapley values offer a fair and justifiable method for evaluating contributions in cooperative game theory.

### 3.2 PROPOSED ERASE-AND-AWAKEN TRAINING STRATEGY

Our objective is to design a training strategy to alleviate the co-adaption problem in SR algorithms, which impedes the generalization capacity, and promote equitable contributions of each channel to predictions. Considering that the prediction in networks relies on the intricate connections among all filters, we introduce the cooperative game theory to formulate this process. For a fully convolutional neural network, denoted as $f_\theta$, it is composed of $L$ layers, each equipped with $n_{l \in \{1,...,L\}}$ filters, and

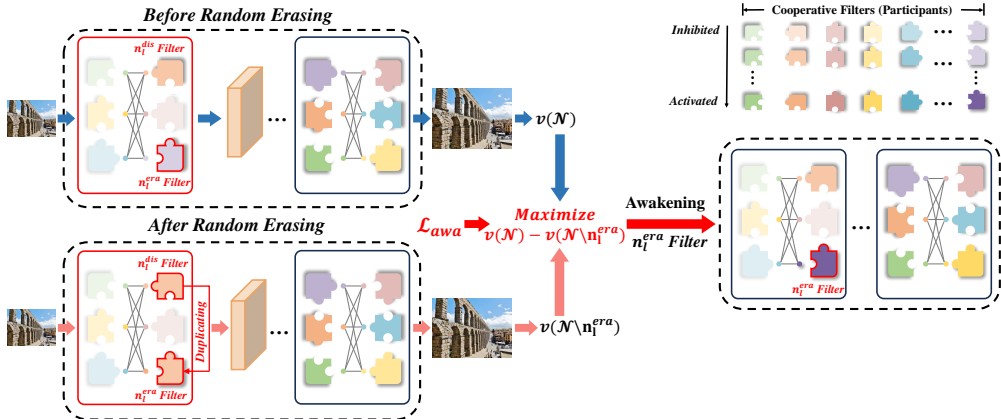

Figure 2: The paradigm of the Erase-and-Awaken Training Strategy (EATS) consists of erasing and awakening steps. **Erasing:** In this step, EATS randomly samples an erased filter and a disruptor filter in a random layer, $n_l^{ear}$ and $n_l^{dis}$ in the $l^{th}$ layer. To assess the contribution of the erased filter, EATS erases $n_l^{era}$ with a duplication of $n_l^{dis}$ and evaluates the performance of networks before and after erasure. **Awakening:** An awakening regularization term, $\mathcal{L}_{awa}$, widens the performance gap before and after erasure, effectively awakening the contribution of the erased filter. Therefore, EATS promotes equitable contributions among filters to predicted high-resolution results and alleviates co-adaption, thereby improving the network's generalization capability.

$n_l^i$ denotes the $i^{th}$ filter in the $l^{th}$ layer. Each filter within the network corresponds to an individual player in the cooperative game model.

**Erasing.** To assess the contribution of each filter, we first randomly erase a filter and then evaluate the performance gap before and after erasure. Instead of directly dropping the filter to be measured, we opt to randomly sample a disruptor filter within the same layer, and employ the sampled filter to erase it. This approach reduces the correlations among channels and promotes feature diversity, as demonstrated in Section 4.3.1. Formally, we define the erased filter as $n_l^{era}$ and the randomly sampled disruptor as $n_l^{dis}$. The process of selecting filter nodes $n_l^{era}$ and $n_l^{dis}$ is:

$$l = \max\{P(i) \in N[0,1]\}, \quad i \in \{1, ..., L\},$$

$$P(n_l^j = n_l^{era}) = \frac{1}{|n_l|}, \quad P(n_l^k = n_l^{dis}) = \frac{1}{|n_l|}, \quad j,k \in \{1, ..., |n_l|\}, \quad j \neq k \tag{2}$$

The networks before and after erasure can be expressed as follows:

$$\{n_1^1, ...\}, \{n_l^1, ...n_l^{era}, ..., n_l^{dis}, ...\}, \{n_L^1, ...\} \rightarrow f_\theta,$$
$$\{n_1^1, ...\}, \{n_1^1, ..., n_l^{dis}, ..., n_l^{dis}, ...\}, \{n_L^1, ...\} \rightarrow f_{\theta'}, \tag{3}$$

where $f_\theta$ denotes the original network and $f_{\theta'}$ represents the network after erasure, $P(\cdot)$ indicates the probability function and $|n_l|$ represents the number of filters in the $l^{th}$ layer.

**Awakening.** To promote the contribution of the erased filter on the prediction, we introduce an awakening regularization term, $\mathcal{L}_{awa}$, aimed at constraining that the predicted high-resolution image of the disrupted network closely approximates a baseline image. In this paper, the baseline is defined as the upsampled image using bicubic interpolation. Given a low-resolution image, $I_{LR}$, an upsampled image via bicubic interpolation, $I_{Bic}$, and the corresponding high-resolution image, $I_{HR}$, the awakening regularization term is defined as:

$$\mathcal{L}_{awa} = ||f_{\theta'}(I_{LR}) - I_{Bic})||_1, \tag{4}$$

where $|| \cdot ||_1$ represents $\ell_1$ norm. The total loss function of the SR network is formulated as: $\mathcal{L} = \mathcal{L}_{ori} + \mathcal{L}_{awa}$, and $\mathcal{L}_{ori} = ||f_\theta(I_{LR}) - I_{HR}||_1$. The awakening regularization term amplifies the performance gap between the original network and the disrupted network, thus enhancing the contribution of the erased filter, $n_l^{era}$, on the prediction.

Since all filters among all layers should be active participants in the cooperative game model and contribute equitably to the prediction, we extend the erase-and-awaken training strategy across all

---

**Algorithm 1:** Erase-and-Awaken Training Strategy

---

**Input:** Paired data with $\{I_{LR}, I_{Bic}, I_{HR}\}$, learning rate $\eta$, updated network $f_\theta$ with $L$ layers.

**Output:** Trained network $f_{\theta^*}$.

**while** *not converge* **do**

> Randomly sample a layer, $l$ from $\{1, \ldots, L\}$ ;
>
> Randomly sample an erased filter, $n_l^{era}$, and a disruptor filter, $n_l^{dis}$, from $l^{th}$ layer by Eq. 2 ;
>
> $f_{\theta'} \leftarrow$ Obtained by Eq. 3 ;
>
> $\mathcal{L}_{awa} \leftarrow ||f_{\theta'}(I_{LR}) - I_{Bic}||_1$ ;
>
> $\mathcal{L}_{ori} \leftarrow ||f_{\theta}(I_{LR}) - I_{HR}||_1$ ;
>
> Update $\theta \leftarrow \theta - \eta\nabla_\theta(\mathcal{L}_{awa} + \mathcal{L}_{ori})$

**end**

---

layers. Specifically, we first randomly select a target layer and then apply the awakening regularization term on the randomly sampled erased and disruptor filters within the target layer. Our main implementation is outlined in Algorithm 1. We also present the analysis in Section 4.3.1 to demonstrate that our EATS fosters equitable contributions among all filters across all layers.

### 3.3 THEORETICAL PROOF IN AWAKENING INHIBITED FILTERS

In Section 3.2, we have introduced the training strategy that conceptualizes all filters within the network as active participants engaged in a collaborative game model. Its primary goal is to promote equitable contributions among all filters, ultimately enhancing overall generalization performance. Here, we leverage the Shapley value from the collaborative game theory to provide theoretical evidence, supporting the effectiveness of EATS in promoting equitable contributions among all filters.

For a fully convolutional neural network, $\mathcal{N}$, the Shapley value of the the $i^{th}$ filter in the $l^{th}$ layer, $n_l^i$, is represented as $\phi_{n_l^i}(\mathcal{M}, v)$, where $\mathcal{M}$ represents the sub-networks within the full network, *i.e.*, $\mathcal{M} \subseteq \mathcal{N}$. To facilitate the proof, we reformulate the Eq. 1 as follows:

$$\phi_{n_l^i}(\mathcal{M}, v) = \frac{1}{(|\mathcal{N}| - 1)!} \sum_{\langle \mathcal{M} \rangle = |\mathcal{N}| - 1} \frac{1}{|\mathcal{N}|} \sum_{\mathcal{S} \subset \mathcal{M}} [v(\mathcal{S} \cup \{n_l^i\}) - v(\mathcal{S})], \qquad (5)$$

where $\langle \mathcal{M} \rangle = |\mathcal{N}| - 1$ represents the set of ordered sequences of length $|\mathcal{N}| - 1$ and $\mathcal{M}$ encompasses all consecutive subsequences of an ordered sequence starting from the first item, including the empty sequence. For example, if we have a set $\{A, B, C\}$, $|\mathcal{M}| = 3$ includes the ordered sequence such as $\{ABC, ACB, BAC, BCA, CAB, CBA\}$. For the sequence $\{ABC\}$, $\mathcal{M} = \{\emptyset, A, AB, ABC\}$, and for the sequence $\{BCA\}$, $\mathcal{M} = \{\emptyset, B, BC, BCA\}$.

We assume that the contribution of $n_l^i$ decreases as the number of filters in $l^{th}$ layer increases (*e.g.*, the contribution of $A$ in $\{ABC\}$ is smaller than that in $\{AB\}$), we have:

$$\begin{aligned}
\phi_{n_l^i}(\mathcal{M}, v) &= \frac{1}{(|\mathcal{N}| - 1)!} \sum_{\langle \mathcal{M} \rangle = |\mathcal{N}| - 1} \frac{1}{|\mathcal{N}|} \sum_{\mathcal{S} \subset \mathcal{M}} [v(\mathcal{S} \cup \{n_l^i\}) - v(\mathcal{S})] \\
&\geq \frac{1}{(|\mathcal{N}| - 1)!} \sum_{\langle \mathcal{M} \rangle = |\mathcal{N}| - 1} \frac{1}{|\mathcal{N}|} \sum_{k=1}^{|\mathcal{N}|} [v(\mathcal{N}) - v(\mathcal{N} \backslash n_l^i)] \\
&\geq v(\mathcal{N}) - v(\mathcal{N} \backslash n_l^i),
\end{aligned} \qquad (6)$$

where $\mathcal{N}$ corresponds to the original network $f_\theta$ and $\mathcal{N} \backslash n_l^i$ corresponds to the model after erasure $f_{\theta'}$. Recall that the awakening regularization term widens the performance gap between $f_\theta$ and $f_{\theta'}$ by effectively constraining the performance of $f_{\theta'}$. As a result, our EATS strategy plays a crucial role in awakening the cooperative relationship among all filters and fostering their equitable contributions to the prediction. While the theoretical proof underscores the fundamental effectiveness of our approach, we delve deeper into its effectiveness in improving generalization capability through extensive analyses, as detailed in Section 4.3.1.

## 4 EXPERIMENTS

### 4.1 EXPERIMENTAL SETTINGS

In the SR community, two predominant settings have gained widespread usage: the single-degradation setting (Timofte et al., 2017) and the multi-degradation setting (Wang et al., 2021; Zhang et al., 2021; Cao et al., 2023). We have opted for the second setting, where overfitting to a specific degradation scenario is no longer a viable option, and the primary challenge revolves around improving the generalization capacity of SR networks.

Following the configuration in Kong et al. (2022), we adopt the high-order degradation modelling (Wang et al., 2021) to conduct the multi-degradation setting. It involves sophisticated combinations of different degradations, including blur, downsampling, noise, and compression. Notably, these combinations are not applied singularly, but rather in multiple iterations to generate multi-faceted degradations. In addition, all factors (*e.g.*, kernels, downsampling scales, noise levels, and compression parameters) are subject to random sampling throughout the training process, where the hyper-parameters are identical to Wang et al. (2021); Kong et al. (2022).

We employ the high-quality images from the DIV2K (Agustsson & Timofte, 2017) dataset for training and the images from Set5 (Bevilacqua et al., 2012), Set14 (Yang et al., 2010), Manga109 (Matsui et al., 2017), Urban100 (Huang et al., 2015), and BSD100 (Martin et al., 2001) for testing. All models in this paper are implemented with PyTorch on NVIDIA GTX 3090 GPUs. We employ an Adam optimizer (Kingma & Ba, 2014) with $\beta_1 = 0.9$, $\beta_2 = 0.999$ to update our model with a batch of 16. The initial learning rate is set to $2 \times 10^{-4}$ and subsequently modulated using the cosine annealing strategy. The patch size of the high-resolution is set to $128 \times 128$.

### 4.2 IMPLEMENTATION DETAILS

We select two representative SR networks, SRResNet Ledig et al. (2017) and RRDB Wang et al. (2018b), consistent with Kong et al. (2022), for our experimental investigations . To validate the efficacy of our erase-and-awaken training strategy, we create distinct variants of the baselines:

- Original: the baseline without any modifications;
- Dropout: incorporating the channel-wise dropout before the last convolution layer, where the drop probability is set to 0.5 and 0.7 for SRResNet and RRDB, respectively, according to the conclusion in Kong et al. (2022);
- EATS: training the baseline model with our Erase-and-Awaken Training Strategy (EATS).

For fair comparisons, we ensure that each competitive baseline and its variants undergo identical training configuration and optimization strategy.

### 4.3 COMPARISON AND ANALYSIS

#### 4.3.1 ANALYSIS OF EATS

**EATS encourages channels to contribute more equally to the prediction, alleviating the co-adapting problem.** To provide further insight into the effectiveness of our EATS, we visualize the feature maps and channel salience maps (CSM) Kong et al. (2022). CSM is a gradient-based attribution analysis method that quantifies the contribution of each channel to the final result. In Fig. 3, we present the feature maps and corresponding CSMs for the output convolutional layer of both SRResNet and SRResNet-EATS. These visualizations demonstrate that integrating our training strategy can equalize the contributions of each channel to the prediction. This balanced contribution is crucial in improving the generalization capacity of the network.

**EATS reduces correlations among channels.** Our EATS randomly erases a filter in the SR network and constrains a performance drop for the erased network. This strategic approach aims to mitigate redundancy in the channel dimension while promoting feature diversity. To illustrate this effect, we conducted an analysis of channel correlations among both shallow and deep features, specifically focusing on the $2^{nd}$ and $14^{th}$ residual blocks of SRResNet and SRResNet-EATS. We utilize cosine similarity as a metric to measure the correlation of all paired channel responses within the same

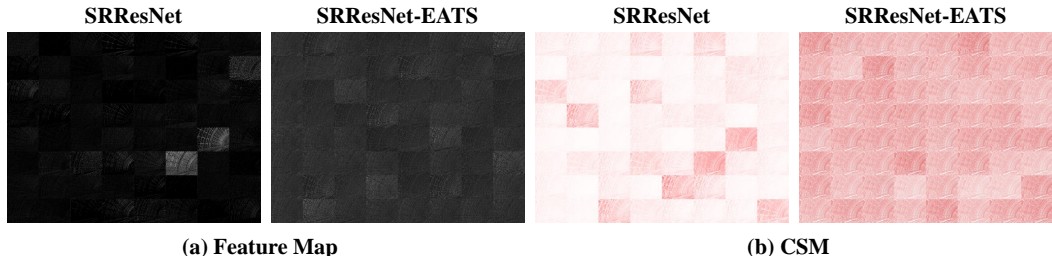

**(a) Feature Map**  **(b) CSM**

Figure 3: The feature maps and CSM of SRResNet and SRResNet-EATS. It indicates that our EATS can promote the baseline to attain equatable channel responses and contributions to the prediction.

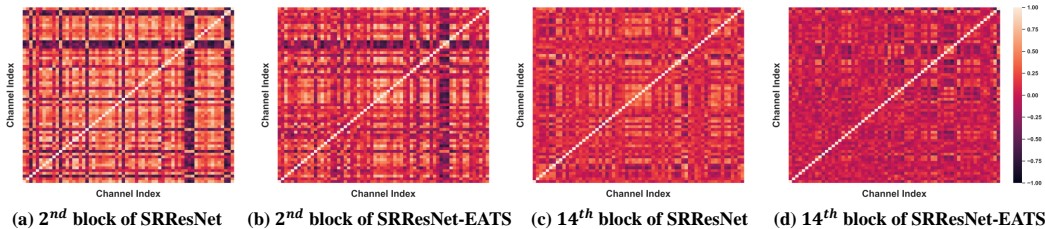

**(a) $2^{nd}$ block of SRResNet** **(b) $2^{nd}$ block of SRResNet-EATS** **(c) $14^{th}$ block of SRResNet** **(d) $14^{th}$ block of SRResNet-EATS**

Figure 4: Cosine similarity among channels of SRResNet and SRResNet-EATS. We select features from the $2^{nd}$ and $14^{th}$ residual blocks as the shallow and deep features. It indicates that integrating with our EATS reduces the correlation among feature channels and increases feature diversity.

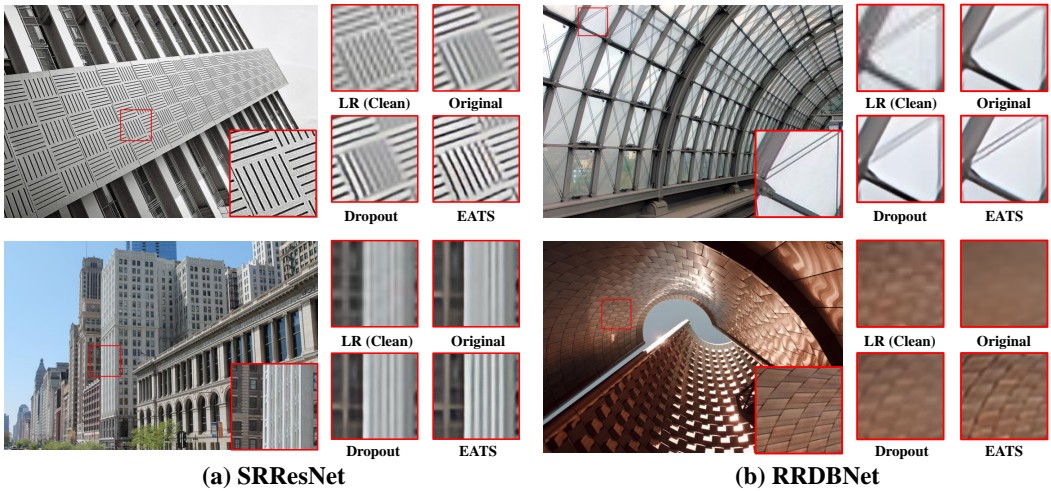

**(a) SRResNet**  **(b) RRDBNet**

Figure 5: Visual comparison of SRResNet Ledig et al. (2017) and RRDBNet Wang et al. (2018b) on the Urban100 Huang et al. (2015) dataset.

layer. The visualization, as shown in Fig. 4 reveals that the channel responses of SRResNet exhibit a higher correlation, indicative of feature redundancy. In contrast, our EATS encourages the baseline to reduce the correlations in feature channels, thereby enhancing feature abundance.

### 4.3.2 RESULTS ON MULTI-DEGRADATION SR

To assess the generalization capacity, we create the complex degradations and their combinations using the high-quality images from the test sets through the data generation pipeline (Wang et al., 2021). Following Kong et al. (2022), we adopt Gaussian blur (kernel size 21 and standard deviation2, indicated by 'b'), bicubic downsampling, Gaussian noise (standard deviation 20, indicated by 'n'), and JPEG compression (quality 50, indicated by 'j') as the testing degradations. Furthermore, we generate complicated degradations formed through the combination of the above components.

Table 1: The PSNR results of SRResNet with $\times 4$ downsampling scales. We apply bicubic, blur, noise, and jpeg to generate the degradation, *e.g.*, clean means only bicubic, b+n means blur $\rightarrow$ bicubic $\rightarrow$ noise. "Improvement" represents the performance difference between the SRResNet-EATS and the original SRResNet.

| Config | Set5 | | | | | | | |
|---|---|---|---|---|---|---|---|---|
| | clean | blur | noise | jpeg | b+n | b+j | n+j | b+n+j |
| Original | 24.9697 | 24.6919 | 21.9275 | 23.7601 | 23.3022 | 23.4550 | 23.2996 | 22.9266 |
| Dropout | 25.6368 | 25.2455 | 21.9388 | 24.0995 | 23.4923 | 23.6830 | 23.5182 | 23.0791 |
| EATS | 25.7532 | 25.3635 | 22.0025 | 24.1728 | 23.5464 | 23.7347 | 23.5546 | 23.1193 |
| Improvement | **+0.7835** | **+0.6716** | **+0.0750** | **+0.4127** | **+0.2442** | **+0.2797** | **+0.2550** | **+0.1927** |
| | Set14 | | | | | | | |
| Original | 22.6507 | 22.5816 | 20.7185 | 21.9326 | 21.9249 | 21.8599 | 21.7380 | 21.5619 |
| Dropout | 23.0847 | 22.9241 | 20.7914 | 22.1585 | 22.0182 | 22.0442 | 21.8873 | 21.6992 |
| EATS | 23.1282 | 22.9352 | 20.8249 | 22.2012 | 22.0533 | 22.0525 | 21.9293 | 21.7322 |
| Improvement | **+0.4775** | **+0.3536** | **+0.1064** | **+0.2686** | **+0.1284** | **+0.1926** | **+0.1913** | **+0.1703** |
| | BSDS100 | | | | | | | |
| Original | 23.0997 | 23.0040 | 21.0617 | 22.5547 | 22.2802 | 22.4339 | 22.3151 | 22.1608 |
| Dropout | 23.4201 | 23.2857 | 21.1229 | 22.7574 | 22.3058 | 22.6000 | 22.4333 | 22.2425 |
| EATS | 23.4684 | 23.3306 | 21.1497 | 22.7820 | 22.3290 | 22.6195 | 22.4610 | 22.2602 |
| Improvement | **+0.3687** | **+0.3266** | **+0.0880** | **+0.2273** | **+0.0488** | **+0.1856** | **+0.1459** | **+0.0994** |
| | Urban100 | | | | | | | |
| Original | 21.2966 | 21.0583 | 19.5806 | 20.7325 | 20.4667 | 20.4127 | 20.5519 | 20.1973 |
| Dropout | 21.6198 | 21.3153 | 19.6532 | 20.9153 | 20.5167 | 20.5494 | 20.6743 | 20.2759 |
| EATS | 21.6097 | 21.3348 | 19.6771 | 20.9109 | 20.5399 | 20.5525 | 20.6899 | 20.2883 |
| Improvement | **+0.3131** | **+0.2765** | **+0.0965** | **+0.1784** | **+0.0732** | **+0.1398** | **+0.1380** | **+0.0910** |
| | Manga109 | | | | | | | |
| Original | 18.6740 | 18.9396 | 18.3484 | 18.4986 | 18.8332 | 18.7030 | 18.4797 | 18.6365 |
| Dropout | 19.0441 | 19.3156 | 18.4341 | 18.7711 | 19.0785 | 18.9471 | 18.7104 | 18.8350 |
| EATS | 19.0727 | 19.3203 | 18.4707 | 18.8026 | 19.0949 | 18.9529 | 18.7504 | 18.8522 |
| Improvement | **+0.3987** | **+0.3807** | **+0.1223** | **+0.3040** | **+0.2617** | **+0.2499** | **+0.2707** | **+0.2157** |

Tables 1 and 2 provide detailed quantitative evaluations of SRResNet and RRDBNet with distinct configurations across the various degradation conditions, demonstrating the effectiveness of our erase-and-awaken training strategy. Although dropout has undeniably enhanced the algorithm's generalization capability, our EATS exhibits remarkable potential for achieving even more substantial improvements. Comparing the results with the baseline model, we observe that EATS leads to remarkable improvements, with the maximum enhancement reaching 0.78 dB for SRResNet and an impressive 0.73 dB for RRDBNet. Furthermore, we compare EATS to dropout, the maximum gap extends to 0.12 dB for SRResNet and 0.21 dB for RRDB. In addition, visualizations in Fig. 5 provide qualitative evidence of EATS's effectiveness. By incorporating our EATS with the original baseline, the super-resolved images achieve more realistic content reconstruction and fine-grained textures without introducing artifacts.

## 5 LIMITATIONS

We will validate the effectiveness of the proposed training strategy on a broader spectrum of low-level tasks, including but not limited to image denoising and image deblurring. As a versatile training strategy, we encourage the exploration of various comprehensive networks integrated with our proposed approach. Our research goes beyond the design of a universal strategy for improving

Table 2: The PSNR results of RRDBNet with ×4 downsampling scale. We apply bicubic, blur, noise, and jpeg to generate the degradation, *e.g.*, clean means only bicubic, b+n means blur → bicubic → noise. "Improvement" represents the performance difference between the RRDBNet-EATS and the original RRDBNet.

| Config | Set5 | | | | | | | |
|---|---|---|---|---|---|---|---|---|
| | clean | blur | noise | jpeg | b+n | b+j | n+j | b+n+j |
| Original | 25.2688 | 25.2776 | 22.2421 | 23.9881 | 23.3838 | 23.6890 | 23.4186 | 22.9922 |
| Dropout | 25.9292 | 25.5746 | 22.7143 | 24.3801 | 23.5175 | 23.9188 | 23.6084 | 23.1050 |
| EATS | 26.0074 | 25.7857 | 22.5926 | 24.4301 | 23.4469 | 23.8805 | 23.6479 | 23.0888 |
| Improvement | **+0.7386** | **+0.5081** | **+0.3505** | **+0.4420** | **+0.0631** | **+0.1915** | **+0.2293** | **+0.0966** |
| | Set14 | | | | | | | |
| Original | 22.9262 | 22.8528 | 20.9357 | 22.1325 | 22.0795 | 21.9733 | 21.9026 | 21.6333 |
| Dropout | 23.2198 | 23.0626 | 21.0197 | 22.3072 | 22.2244 | 22.0923 | 22.0669 | 21.7556 |
| EATS | 23.4922 | 23.2615 | 21.2980 | 22.4989 | 22.1894 | 22.1938 | 22.1133 | 21.7674 |
| Improvement | **+0.5660** | **+0.4087** | **+0.3623** | **+0.3664** | **+0.1099** | **+0.2205** | **+0.2107** | **+0.1341** |
| | BSDS100 | | | | | | | |
| Original | 23.3654 | 23.3880 | 21.2915 | 22.7456 | 22.3154 | 22.6020 | 22.4485 | 22.2281 |
| Dropout | 23.5886 | 23.5616 | 21.4746 | 22.8828 | 22.3895 | 22.7058 | 22.5420 | 22.2849 |
| EATS | 23.6963 | 23.6864 | 21.5848 | 22.9928 | 22.4043 | 22.7814 | 22.5743 | 22.3039 |
| Improvement | **+0.3309** | **+0.2984** | **+0.2933** | **+0.2472** | **+0.0889** | **+0.1794** | **+0.1258** | **+0.0758** |
| | Urban100 | | | | | | | |
| Original | 21.5738 | 21.4637 | 19.5775 | 20.9649 | 20.4764 | 20.5970 | 20.7884 | 20.3406 |
| Dropout | 21.8104 | 21.6082 | 19.6290 | 21.1218 | 20.5096 | 20.6922 | 20.8822 | 20.3855 |
| EATS | 21.9979 | 21.6462 | 19.8111 | 21.2543 | 20.5088 | 20.7175 | 20.9688 | 20.4000 |
| Improvement | **+0.4241** | **+0.1825** | **+0.2336** | **+0.2894** | **+0.0324** | **+0.1205** | **+0.1804** | **+0.0594** |
| | Manga109 | | | | | | | |
| Original | 18.6101 | 18.8597 | 18.4166 | 18.5256 | 19.0068 | 18.6980 | 18.5759 | 18.6658 |
| Dropout | 18.9249 | 19.0455 | 18.6042 | 18.7816 | 19.1874 | 18.8970 | 18.7588 | 18.7510 |
| EATS | 18.9220 | 19.0082 | 18.6622 | 18.8107 | 19.1926 | 18.9026 | 18.7213 | 18.7592 |
| Improvement | **+0.3119** | **+0.1485** | **+0.2456** | **+0.2851** | **+0.1858** | **+0.2046** | **+0.1454** | **+0.0934** |

the generalization performance of existing networks. It aims to introduce an alternative perspective, grounded in cooperative games within neural networks, for advancing generalization capacity. Therefore, our ongoing efforts will be dedicated to demonstrating the effectiveness of this approach across a wider range of computer vision tasks.

## 6 CONCLUSION

In this paper, we have introduced cooperative game theory to enhance the generalization capacity of image super-resolution algorithms in real-world scenarios, and proposed an Erase-and-Awaken Training Strategy (EATS). It treats all neurons within the network as active participants in a collaborative relationship, collectively determining the final prediction. Our EATS strategy effectively awakens previously suppressed neurons that hinder generalization and promotes equitable contributions among all neurons, thus alleviating effectively co-adaption problem and improving generalization performance. We provide theoretical proof of its effectiveness in promoting the contribution of each neuron to predictions. Notably, our approach can seamlessly integrate with existing networks, reinforcing their ability to generalize across unforeseen scenarios. Extensive experiments across various unseen datasets with distinct degradations consistently demonstrate the substantial performance gains achieved by incorporating our strategy.

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

APPENDIX

- We further evaluate the effectiveness of our EATS on super-resolution with $\times 2$ scaling.
- We evaluate the effectiveness of our EATS on another attention-based algorithm.
- We discuss the training costs incurred by our proposed EATS.
- We validate the convergence of the awakening regularization in our EATS.
- We conduct the ablation studies to investigate the impact of varying layer ratios in the erase-and-awaken training strategy on SRResNet.
- We conduct the ablation studies to investigate the impact of the number of involved filters in each training iteration on SRResNet.
- We analyze the effectiveness of our EATS on reducing correlations among channels on more layers within SRResNet and RRDBNet.
- We analyze the effectiveness of our EATS on alleviating co-adaption phenomenon in both shallow and deep layers.
- We present more visualization results on different degradation configurations.

## A    GENERALIZATION PERFORMANCE ON X2 SCALING

Tab. 3 and 4 provide quantitative evaluations of SRResNet and RRDBNet on ×2 scaling, which further demonstrates the effectiveness of our erase-and-awaken training strategy. Comparing the results with the baseline model, we observe that EATS leads to remarkable improvements, with the maximum enhancement reaching 0.53 dB for SRResNet and an impressive 0.66 dB for RRDBNet. Furthermore, we comare EATS to dropout, the maximum gap extends to 0.15 dB for SRResNet and 0.12 dB for RRDB.

Table 3: The PSNR results of SRResNet with ×2. We apply bicubic, blur, noise and jpeg to generate the degradation, *e.g.*, clean means only bicubic, b+n means blur → bicubic → noise. "Improvement" represents that the performance enhancement of SRResNet-EATS compared to the SRResNet.

| Config | Set5 | | | | | | | |
|---|---|---|---|---|---|---|---|---|
| | clean | blur | noise | jpeg | b+n | b+j | n+j | b+n+j |
| Original | 26.8084 | 26.8084 | 24.1133 | 26.8350 | 25.0767 | 25.8566 | 26.1467 | 24.9306 |
| Dropout | 27.2030 | 27.2030 | 24.2231 | 27.1348 | 25.1975 | 26.1130 | 26.4176 | 25.1576 |
| EATS | 27.3422 | 27.3422 | 24.2282 | 27.1906 | 25.2221 | 26.1967 | 26.4641 | 25.1835 |
| Improvement | **+0.5338** | **+0.5338** | **+0.1149** | **+0.3556** | **+0.1454** | **+0.3401** | **+0.3174** | **+0.2529** |
| | Set14 | | | | | | | |
| Original | 25.7810 | 24.7398 | 22.8671 | 24.9982 | 23.5337 | 23.9898 | 24.5282 | 23.4352 |
| Dropout | 26.0030 | 24.8413 | 22.9801 | 25.1041 | 23.5441 | 24.0763 | 24.6533 | 23.5380 |
| EATS | 26.0541 | 24.9154 | 22.9606 | 25.1198 | 23.5664 | 24.1157 | 24.6745 | 23.5584 |
| Improvement | **+0.2731** | **+0.1756** | **+0.0935** | **+0.1216** | **+0.0327** | **+0.1259** | **+0.1463** | **+0.1232** |
| | BSDS100 | | | | | | | |
| Original | 25.4314 | 24.6299 | 22.6302 | 24.8488 | 23.4511 | 24.1568 | 24.4636 | 23.6270 |
| Dropout | 25.5721 | 24.7310 | 22.7676 | 24.9370 | 23.5241 | 24.2373 | 24.5331 | 23.7019 |
| EATS | 25.5592 | 24.7149 | 22.7614 | 24.9195 | 23.5120 | 24.2211 | 24.5331 | 23.6943 |
| Improvement | **+0.1278** | **+0.0850** | **+0.1312** | **+0.0707** | **+0.0609** | **+0.0643** | **+0.0695** | **+0.0673** |
| | Urban100 | | | | | | | |
| Original | 24.1966 | 22.6617 | 21.7665 | 23.4633 | 21.7126 | 22.0602 | 23.1523 | 21.6205 |
| Dropout | 24.4202 | 22.7573 | 21.8302 | 23.6326 | 21.7265 | 22.1770 | 23.3069 | 21.7268 |
| EATS | 24.4522 | 22.8045 | 21.8139 | 23.6467 | 21.7413 | 22.2080 | 23.3266 | 21.7425 |
| Improvement | **+0.2556** | **+0.1428** | **+0.0474** | **+0.1834** | **+0.0287** | **+0.1478** | **+0.1743** | **+0.1220** |
| | Manga100 | | | | | | | |
| Original | 25.4198 | 24.6002 | 23.2008 | 24.6422 | 23.4303 | 23.7406 | 24.3533 | 23.1404 |
| Dropout | 25.9607 | 24.7978 | 23.3060 | 24.9350 | 23.5122 | 23.8499 | 24.6198 | 23.2815 |
| EATS | 25.9105 | 24.8264 | 23.2846 | 24.9009 | 23.5090 | 23.8747 | 24.6052 | 23.2811 |
| Improvement | **+0.4907** | **+0.2262** | **+0.0838** | **+0.2587** | **+0.0787** | **+0.1341** | **+0.2519** | **+0.1407** |

Table 4: The PSNR results of RRDBNet with ×2. We apply bicubic, blur, noise and jpeg to generate the degradation, *e.g.*, clean means only bicubic, b+n means blur → bicubic → noise. "Improvement" represents that the performance enhancement of RRDBNet-EATS compared to the RRDBNet.

| Config | Set5 | | | | | | | |
|---|---|---|---|---|---|---|---|---|
| | clean | blur | noise | jpeg | b+n | b+j | n+j | b+n+j |
| Original | 28.1476 | 27.3172 | 25.2403 | 27.1145 | 24.4877 | 26.1389 | 26.4795 | 25.2270 |
| Dropout | 28.7062 | 28.1439 | 25.5222 | 27.4045 | 25.9662 | 26.5147 | 26.7263 | 25.5055 |
| EATS | 28.8104 | 28.2165 | 25.6271 | 27.5202 | 25.9556 | 26.5562 | 26.7342 | 25.4417 |
| Improvement | **+0.6628** | **+0.8993** | **+0.3868** | **+0.4057** | **+1.4679** | **+0.4173** | **+0.2547** | **+0.2147** |
| | Set14 | | | | | | | |
| Original | 24.7354 | 24.2766 | 23.1168 | 24.2496 | 23.1845 | 23.5462 | 23.8335 | 22.9599 |
| Dropout | 24.9120 | 24.5681 | 23.2643 | 24.3428 | 23.3876 | 23.7963 | 23.9163 | 23.1106 |
| EATS | 24.9919 | 24.5677 | 23.3152 | 24.4157 | 23.3765 | 23.7612 | 23.9477 | 23.0522 |
| Improvement | **+0.2565** | **+0.2911** | **+0.1984** | **+0.1661** | **+0.1920** | **+0.2150** | **+0.1142** | **+0.0923** |
| | BSDS100 | | | | | | | |
| Original | 24.4175 | 24.2773 | 23.1084 | 24.0722 | 23.3309 | 23.7638 | 23.4950 | 23.2252 |
| Dropout | 24.5992 | 24.4672 | 23.2565 | 24.1994 | 23.4885 | 23.8946 | 23.8661 | 23.3335 |
| EATS | 24.5628 | 24.4898 | 23.2679 | 24.1741 | 23.4937 | 23.8912 | 23.8320 | 23.3250 |
| Improvement | **+0.1453** | **+0.2125** | **+0.1595** | **+0.1019** | **+0.1628** | **+0.1274** | **+0.3370** | **+0.0998** |
| | Urban100 | | | | | | | |
| Original | 24.3659 | 23.1373 | 22.5218 | 23.7493 | 21.9898 | 22.4042 | 23.3622 | 21.8094 |
| Dropout | 24.6209 | 23.3539 | 22.6854 | 23.9605 | 22.1361 | 22.5125 | 23.5329 | 21.9296 |
| EATS | 24.5982 | 23.4233 | 22.7519 | 23.9508 | 22.1754 | 22.5445 | 23.5341 | 21.9513 |
| Improvement | **+0.2323** | **+0.2860** | **+0.2301** | **+0.2015** | **+0.1856** | **0.1403** | **+0.1719** | **+0.1419** |
| | Manga100 | | | | | | | |
| Original | 19.6360 | 20.2221 | 19.9389 | 19.6657 | 20.3215 | 20.1687 | 19.7929 | 20.2302 |
| Dropout | 19.7362 | 20.5398 | 20.0121 | 19.7291 | 20.5115 | 20.4014 | 19.8055 | 20.3722 |
| EATS | 19.7097 | 20.4536 | 20.0801 | 19.7278 | 20.4450 | 20.3553 | 19.8096 | 20.3086 |
| Improvement | **+0.0737** | **+0.2315** | **+0.1412** | **+0.0621** | **+0.1235** | **+0.1866** | **+0.0167** | **+0.0784** |

# B PERFORMANCE ON ATTENTION-BASED APPROACH

we have evaluated the effectiveness of our proposed EATS using a lightweight variant of HAN Niu et al. (2020) with x4 scaling. This lightweight version comprises approximately 1/20 of the parameters found in the original network. To maintain consistency with the experimental settings outlined in the manuscript, we have re-trained both the lightweight HAN and HAN-EATS, incorporating high-order degradation modeling. The quantitative results are presented in Table 5.

Table 5: The PSNR results of HAN with ×4. We apply bicubic, blur, noise and jpeg to generate the degradation, *e.g.*, clean means only bicubic, b+n means blur → bicubic → noise. "Improvement" represents that the performance enhancement of HAN-EATS compared to the HAN.

| Config | Set5 | | | | | | | |
|---|---|---|---|---|---|---|---|---|
| | clean | blur | noise | jpeg | b+n | b+j | n+j | b+n+j |
| Original | 25.4834 | 25.4207 | 22.5964 | 24.1479 | 23.7007 | 23.8615 | 23.5668 | 23.1275 |
| EATS | 26.1176 | 25.8898 | 22.8330 | 24.4354 | 23.8923 | 23.9615 | 23.6703 | 23.1699 |
| Improvement | **0.6342** | **0.4691** | **0.2366** | **0.2875** | **0.1916** | **0.1000** | **0.1035** | **0.0424** |
| | Set14 | | | | | | | |
| Original | 23.0610 | 23.1144 | 21.3902 | 22.2880 | 22.3199 | 22.2746 | 22.0212 | 21.7860 |
| EATS | 23.4764 | 23.3560 | 21.5058 | 22.4861 | 22.3971 | 22.2772 | 22.1089 | 21.7872 |
| Improvement | **0.4154** | **0.2416** | **0.1156** | **0.1981** | **0.0772** | **0.0026** | **0.0877** | **0.0012** |
| | BSDS100 | | | | | | | |
| Original | 23.4619 | 23.4352 | 21.7799 | 22.8227 | 22.5801 | 22.7466 | 22.4817 | 22.2781 |
| EATS | 23.6903 | 23.5756 | 21.8599 | 22.9232 | 22.6394 | 22.7573 | 22.5339 | 22.2937 |
| Improvement | **0.2284** | **0.1404** | **0.0800** | **0.1005** | **0.0593** | **0.0107** | **0.0522** | **0.0156** |
| | Urban100 | | | | | | | |
| Original | 21.6242 | 21.5168 | 20.2169 | 21.0329 | 20.7608 | 20.7810 | 20.7895 | 20.3836 |
| EATS | 21.9152 | 21.7400 | 20.2708 | 21.1839 | 20.7677 | 20.8224 | 20.8381 | 20.3987 |
| Improvement | **0.2910** | **0.2232** | **0.0539** | **0.1510** | **0.0069** | **0.0414** | **0.0486** | **0.0151** |
| | Manga100 | | | | | | | |
| Original | 18.9412 | 19.2265 | 18.6995 | 18.7634 | 19.1281 | 19.0087 | 18.7372 | 18.7624 |
| EATS | 19.3123 | 19.5149 | 18.8154 | 19.0205 | 19.2600 | 19.1412 | 18.8486 | 18.9039 |
| Improvement | **0.3711** | **0.2884** | **0.1159** | **0.2571** | **0.1319** | **0.1325** | **0.1114** | **0.1415** |

## C    TRAINING COSTS INCURRED BY OUR EATS

Firstly, it's essential to emphasize that our proposed Erasing and Awakening Training Scheme (EATS) does not introduce additional training parameters. Within a training iteration, we conduct a forward propagation of the original network, $f_\theta$, calculating the original loss function, $\mathcal{L}_{ori}$. Subsequently, our EATS involves randomly replacing an erased filter with a disrupted filter based on the original network, resulting in a modified network after erasure, $f_{\theta'}$ ). We then calculate the awakening regularization, $\mathcal{L}_{awa}$, through a second forward propagation. The network is update based on the combination of these two loss functions. Therefore, it's crucial to note that our EATS only requires two forward propagations, as opposed to employing two separate neural networks.

Secondly, the two forward propagations in our training paradigm marginally increase training times, approximately by 1.4 times compared to the corresponding original networks. This increment is deemed acceptable, especially considering the significant improvement in generalization capability.

Lastly, it's important to highlight that our training paradigm is exclusively applied during the training process and does not impose any additional computational burden during the inference phase.

## D    CONVERGENCE OF THE AWAKENING REGULARIZATION

We present the visualizations of the original loss, $\mathcal{L}_{ori}$, and our awakening regularization, $\mathcal{L}_{awa}$, during the training process of SRResNet-EATS and RRDBNet-EATS. Through the plots presented in Fig. 6, it is evident that the awakening regularization progressively decreases and converges with the training iteration.

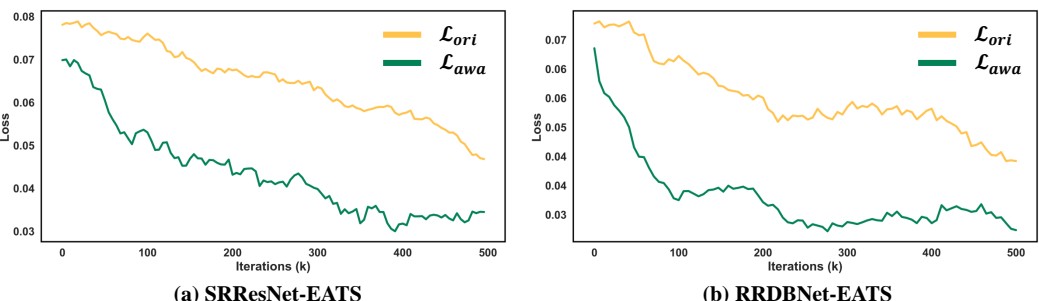

(a) **SRResNet-EATS**                    (b) **RRDBNet-EATS**

Figure 6: The curves of the original loss and the awakening regularization of SRResNet-EATS and RRDBNet-EATS during the training process..

# E ABLATION STUDIES ABOUT THE RATION OF INVOLVED LAYERS

Table 6: Ablation studies about the impact of varying layer ratios involved in the erase-and-awaken training strategy on RRDBNet with × 4 scaling. We apply bicubic, blur, noise and jpeg to generate the degradation, *e.g.*, clean means only bicubic, b+n means blur → bicubic → noise.

| Config | Set5 | | | | | | | |
|---|---|---|---|---|---|---|---|---|
| | clean | blur | noise | jpeg | b+n | b+j | n+j | b+n+j |
| Original | 25.2688 | 25.2776 | 22.2421 | 23.9881 | 23.3838 | 23.6890 | 23.4186 | 22.9922 |
| EATS-20% | 25.4705 | 25.4355 | 22.0787 | 24.0810 | 23.5124 | 23.7079 | 23.4250 | 22.9718 |
| EATS-40% | 25.6724 | 25.7932 | 22.1805 | 24.2320 | 23.6813 | 23.8129 | 23.5765 | 23.0717 |
| EATS-60% | 25.7675 | 25.7484 | 22.3898 | 24.2352 | 23.3675 | 23.8208 | 23.5615 | 23.0379 |
| EATS-80% | 25.8453 | 25.9380 | 22.4460 | 24.2615 | 23.5560 | 23.8024 | 23.5686 | 22.9833 |
| EATS | 26.0074 | 25.7857 | 22.5926 | 24.4301 | 23.4469 | 23.8805 | 23.6479 | 23.0888 |
| | Set14 | | | | | | | |
| Original | 22.9262 | 22.8528 | 20.9357 | 22.1325 | 22.0795 | 21.9733 | 21.9026 | 21.6333 |
| EATS-20% | 23.0829 | 23.0253 | 21.0489 | 22.2179 | 22.2532 | 22.0621 | 21.9563 | 21.6807 |
| EATS-40% | 23.2977 | 23.2232 | 21.1205 | 22.3883 | 22.3323 | 22.1927 | 22.0753 | 21.7727 |
| EATS-60% | 23.3332 | 23.2649 | 21.2404 | 22.3337 | 22.1578 | 22.1454 | 22.0184 | 21.7227 |
| EATS-80% | 23.3325 | 23.2644 | 21.2415 | 22.3583 | 22.2633 | 22.1002 | 22.0565 | 21.5745 |
| EATS | 23.4922 | 23.2615 | 21.2980 | 22.4989 | 22.1894 | 22.1938 | 22.1133 | 21.7674 |
| | BSDS100 | | | | | | | |
| Original | 23.3654 | 23.3880 | 21.2915 | 22.7456 | 22.3154 | 22.6020 | 22.4485 | 22.2281 |
| EATS-20% | 23.4795 | 23.5016 | 21.3462 | 22.8051 | 22.4628 | 22.6454 | 22.4641 | 22.2288 |
| EATS-40% | 23.5306 | 23.5857 | 21.4433 | 22.8932 | 22.5713 | 22.7332 | 22.5348 | 22.2917 |
| EATS-60% | 23.6355 | 23.6796 | 21.4363 | 22.9088 | 22.3937 | 22.7411 | 22.5395 | 22.2907 |
| EATS-80% | 23.6380 | 23.7344 | 21.5845 | 22.9338 | 22.5199 | 22.7489 | 22.5507 | 22.2736 |
| EATS | 23.6963 | 23.6864 | 21.5848 | 22.9928 | 22.4043 | 22.7814 | 22.5743 | 22.3039 |
| | Urban100 | | | | | | | |
| Original | 21.5738 | 21.4637 | 19.5775 | 20.9649 | 20.4764 | 20.5970 | 20.7884 | 20.3406 |
| EATS-20% | 21.7343 | 21.5404 | 19.6680 | 21.0606 | 20.5837 | 20.6305 | 20.8302 | 20.3423 |
| EATS-40% | 21.7314 | 21.6119 | 19.7038 | 21.0855 | 20.6532 | 20.6820 | 20.8592 | 20.3655 |
| EATS-60% | 21.8647 | 21.7017 | 19.7688 | 21.1319 | 20.5048 | 20.6927 | 20.8826 | 20.3864 |
| EATS-80% | 21.9305 | 21.7670 | 19.7500 | 21.1860 | 20.6358 | 20.6939 | 20.9345 | 20.3794 |
| EATS | 21.9979 | 21.6462 | 19.8111 | 21.2543 | 20.5088 | 20.7175 | 20.9688 | 20.4000 |
| | Manga100 | | | | | | | |
| Original | 18.6101 | 18.8597 | 18.4166 | 18.5256 | 19.0068 | 18.6980 | 18.5759 | 18.6658 |
| EATS-20% | 18.7687 | 18.9854 | 18.5131 | 18.6210 | 19.1806 | 18.7449 | 18.6311 | 18.7002 |
| EATS-40% | 18.7363 | 18.9413 | 18.5269 | 18.6576 | 19.2270 | 18.8488 | 18.6335 | 18.7278 |
| EATS-60% | 18.7884 | 19.0606 | 18.5614 | 18.6790 | 19.2359 | 18.8396 | 18.6443 | 18.7303 |
| EATS-80% | 18.7912 | 18.9745 | 18.6295 | 18.6941 | 19.2364 | 18.8263 | 18.6835 | 18.7262 |
| EATS | 19.9220 | 19.0082 | 18.6622 | 18.8107 | 19.1926 | 18.9026 | 18.7213 | 18.7592 |

# F ABLATION STUDIES ABOUT THE NUMBER OF INVOLVED FILTERS IN EACH TRAINING ITERATION

Our experiments explored configurations with 1 (default), 3, and 5 filters engaged in each iteration. Compared to the default of 1 filter, involving 3 filters achieved only a slight improvement. This decline may be attributed to increased randomness and instability during the training process.

Table 7: Ablation studies about the number of involved filters in each training iteration on RRDBNet with $\times$ 4 scaling. We apply bicubic, blur, noise and jpeg to generate the degradation, *e.g.*, clean means only bicubic, b+n means blur $\rightarrow$ bicubic $\rightarrow$ noise.

| Config | Set5 | | | | | | | |
|---|---|---|---|---|---|---|---|---|
| | clean | blur | noise | jpeg | b+n | b+j | n+j | b+n+j |
| Original | 25.2688 | 25.2776 | 22.2421 | 23.9881 | 23.3838 | 23.6890 | 23.4186 | 22.9922 |
| EATS-1filter (default) | 26.0074 | 25.7857 | 22.5926 | 24.4301 | 23.4469 | 23.8805 | 23.6479 | 23.0888 |
| EATS-3filter | 26.0538 | 25.9900 | 22.5482 | 24.4395 | 23.6294 | 23.9155 | 23.5702 | 23.0095 |
| EATS-5filter | 25.8929 | 25.6534 | 22.4461 | 24.2632 | 23.5130 | 23.8032 | 23.5774 | 23.0124 |
| | Set14 | | | | | | | |
| Original | 22.9262 | 22.8528 | 20.9357 | 22.1325 | 22.0795 | 21.9733 | 21.9026 | 21.6333 |
| EATS-1filter (default) | 23.4922 | 23.2615 | 21.2980 | 22.4989 | 22.1894 | 22.1938 | 22.1133 | 21.7674 |
| EATS-3filter | 23.6219 | 23.3394 | 21.3617 | 22.5045 | 22.3048 | 22.1650 | 22.0960 | 21.7275 |
| EATS-5filter | 23.4872 | 23.1491 | 21.2607 | 22.4014 | 22.0382 | 22.1274 | 22.0476 | 21.7356 |
| | BSDS100 | | | | | | | |
| Original | 23.3654 | 23.3880 | 21.2915 | 22.7456 | 22.3154 | 22.6020 | 22.4485 | 22.2281 |
| EATS-1filter (default) | 23.6963 | 23.6864 | 21.5848 | 22.9928 | 22.4043 | 22.7814 | 22.5743 | 22.3039 |
| EATS-3filter | 23.7743 | 23.7075 | 21.7250 | 23.0244 | 22.5536 | 22.8058 | 22.5950 | 22.3234 |
| EATS-5filter | 23.5557 | 23.5699 | 21.4584 | 22.9784 | 22.5033 | 22.7022 | 22.5667 | 22.3020 |
| | Urban100 | | | | | | | |
| Original | 21.5738 | 21.4637 | 19.5775 | 20.9649 | 20.4764 | 20.5970 | 20.7884 | 20.3406 |
| EATS-1filter (default) | 21.9979 | 21.6462 | 19.8111 | 21.2543 | 20.5088 | 20.7175 | 20.9688 | 20.4000 |
| EATS-3filter | 21.9795 | 21.7051 | 19.9301 | 21.2136 | 20.6727 | 20.6532 | 20.9733 | 20.3649 |
| EATS-5filter | 21.8009 | 21.5756 | 19.7940 | 21.1980 | 20.4379 | 20.6861 | 20.9068 | 20.3656 |
| | Manga100 | | | | | | | |
| Original | 18.6101 | 18.8597 | 18.4166 | 18.5256 | 19.0068 | 18.6980 | 18.5759 | 18.6658 |
| EATS-1filter (default) | 19.9220 | 19.0082 | 18.6622 | 18.8107 | 19.1926 | 18.9026 | 18.7213 | 18.7592 |
| EATS-3filter | 19.9446 | 19.2506 | 18.8891 | 18.8846 | 19.2921 | 19.9487 | 18.7675 | 18.7990 |
| EATS-5filter | 19.8026 | 18.9324 | 18.6103 | 18.8145 | 19.1652 | 18.8039 | 18.7179 | 18.7605 |

## G  MORE VISUALIZATIONS ABOUT THE FEATURE CORRELATIONS

We visualize more channel correlation matrix on $2^{nd}$, $8^{th}$, $14^{th}$ and output layer wihtin the SRRes-Net and RRDBNet in Fig. 7 and Fig. 8, respectively. We utilize cosine similarity as a metric to measure the correlation of all paired channel responses within the same layer. The visualization reveals that the channel responses of baseline models exhibit higher correlation, indicative of feature redundancy. In contrast, our EATS encourages the baselines to reduce the correlations in feature channels, thereby enhancing feature abundance.

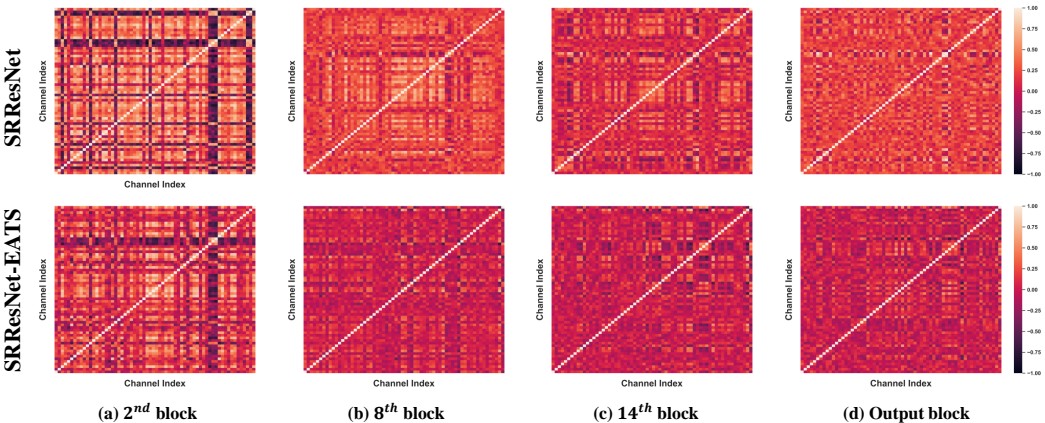

(a) $2^{nd}$ block      (b) $8^{th}$ block      (c) $14^{th}$ block      (d) Output block

Figure 7: Cosine similarity among channels of SRResNet and SRResNet-EATS. We select features from the $2^{nd}$, $8^{th}$, $14^{th}$ and output blocks as the shallow and deep features. It indicates that integrating with our EATS reduces the correlation among feature channels and increases feature diversity.

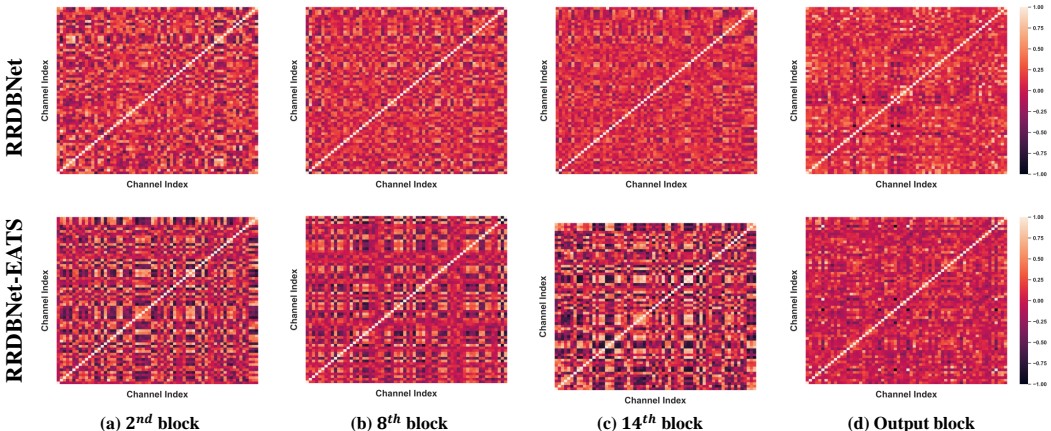

(a) $2^{nd}$ block      (b) $8^{th}$ block      (c) $14^{th}$ block      (d) Output block

Figure 8: Cosine similarity among channels of RRDBNet and RRDBNet-EATS. We select features from the $2^{nd}$, $8^{th}$, $14^{th}$ and output blocks as the shallow and deep features. It indicates that integrating with our EATS reduces the correlation among feature channels and increases feature diversity.

# H ALLEVIATING CO-ADAPTION IN BOTH SHALLOW AND DEEP LAYERS.

The conclusions draw from the previous work Kong et al. (2022) indicates that employing dropout only at the last convolutional layer improves generalization performance, while even causing drop in performance at other layers. Therefore, we conducted a statistical analysis of channel responses in the $2^{nd}$ and last output layers of SRResNet Ledig et al. (2017), SRResNet with dropout, and SRResNet-EATS (all trained on the Five5K Bychkovsky et al. (2011) dataset) across 100 randomly sampled unseen images. The results, as shown in Fig. 9, indicate that while dropout operation can alleviate the co-adaptation Hinton et al. (2012) phenomenon in the last layer, it fails to address the issue in the shallow layers. In contrast, incorporating our training strategy effectively mitigates co-adaptation and achieves equitable channel responses in both shallow and deep layers.

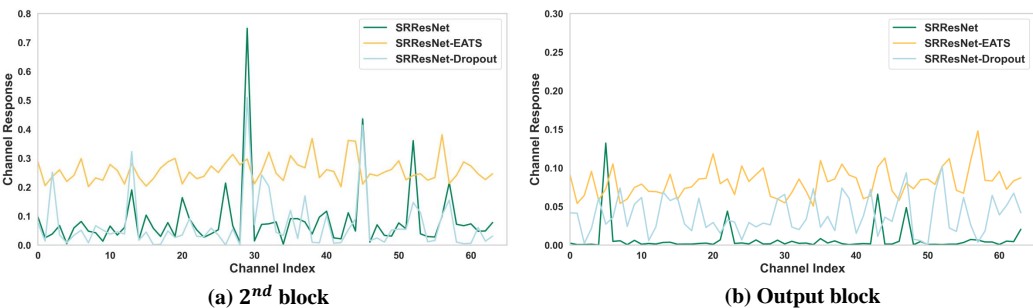

(a) $2^{nd}$ block  (b) Output block

Figure 9: The Erase-and-Awaken Training Strategy (EATS) mitigates the co-adaption problem in both shallow and deep layers, $2^{th}$ and output block. We analyze channel responses in unseen scenarios, averaging channel responses from the $2^{nd}$ block in SRResNet (trained on Five5K) across 100 randomly sampled unseen images. It reveals the co-adaption phenomenon (Hinton et al., 2012), where a few channels are highly activated while others are inhibited. As the dropout operation is strategically applied (only the output layer) Kong et al. (2022), its effectiveness is limited to mitigating co-adaptation in deeper layers. Conversely, the integration of EATS with SRResNet yields balanced and activated channel responses in both shallow and deep layers.

# I MORE VISUALIZATIONS

In this section, we provide additional qualitative results on different degradations to show the effectiveness of our EATS (see Fig. 10 to Fig.17). Following consistent configurations with the manuscript, we adopt Gaussian blur (kernel size 21 and standard deviation2, indicated by "b"), bicubic downsampling, Gaussian noise (standard deviation 20, indicated by "n"), and JPEG compression (quality 50, indicated by "j") as the testing degradations. In addition to the single degradation, we synthesize the complicated mixed degradations with the second degradation process Wang et al. (2021).

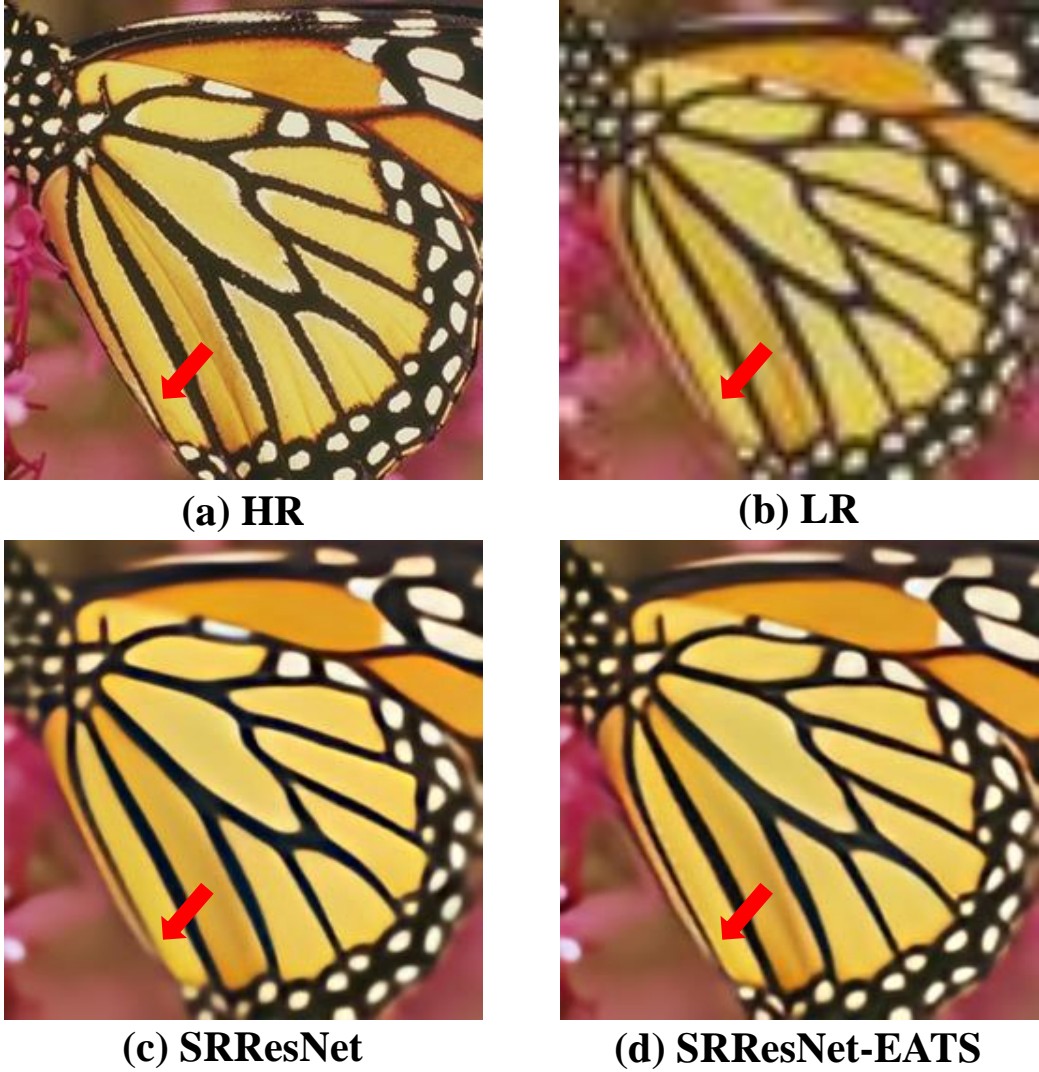

Figure 10: Visual comparison of SRResNet Ledig et al. (2017) on the clean-Set5 Bevilacqua et al. (2012).

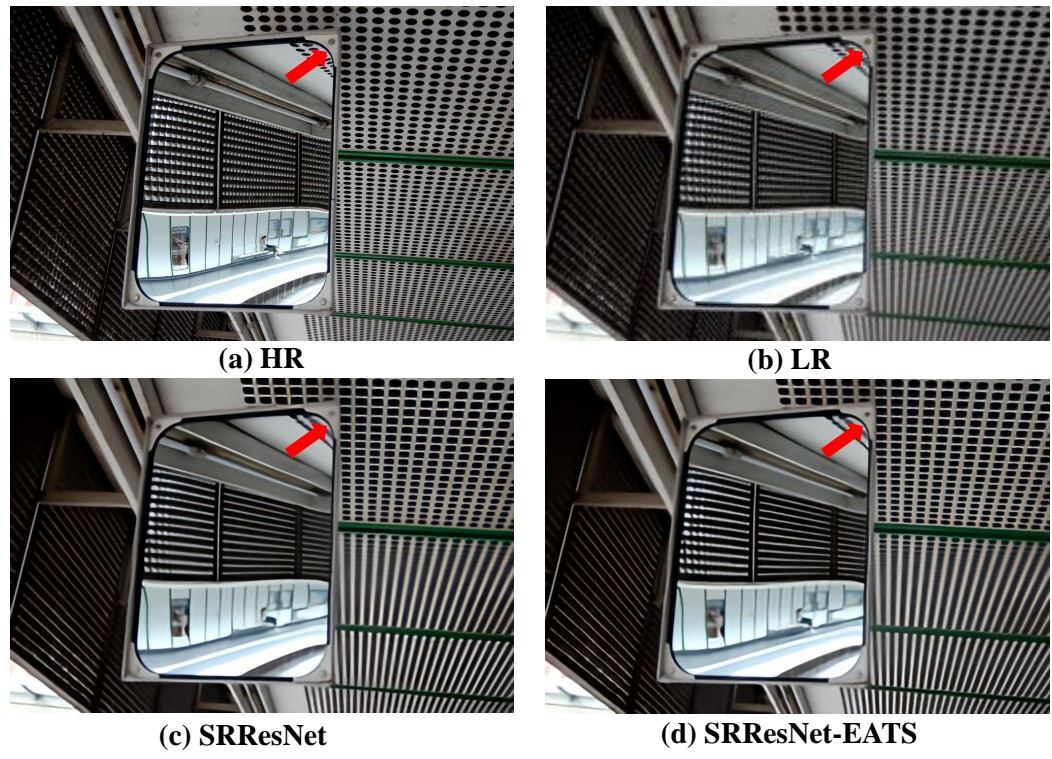

Figure 11: Visual comparison of SRResNet Ledig et al. (2017) on the blur-Urban100 Huang et al. (2015).

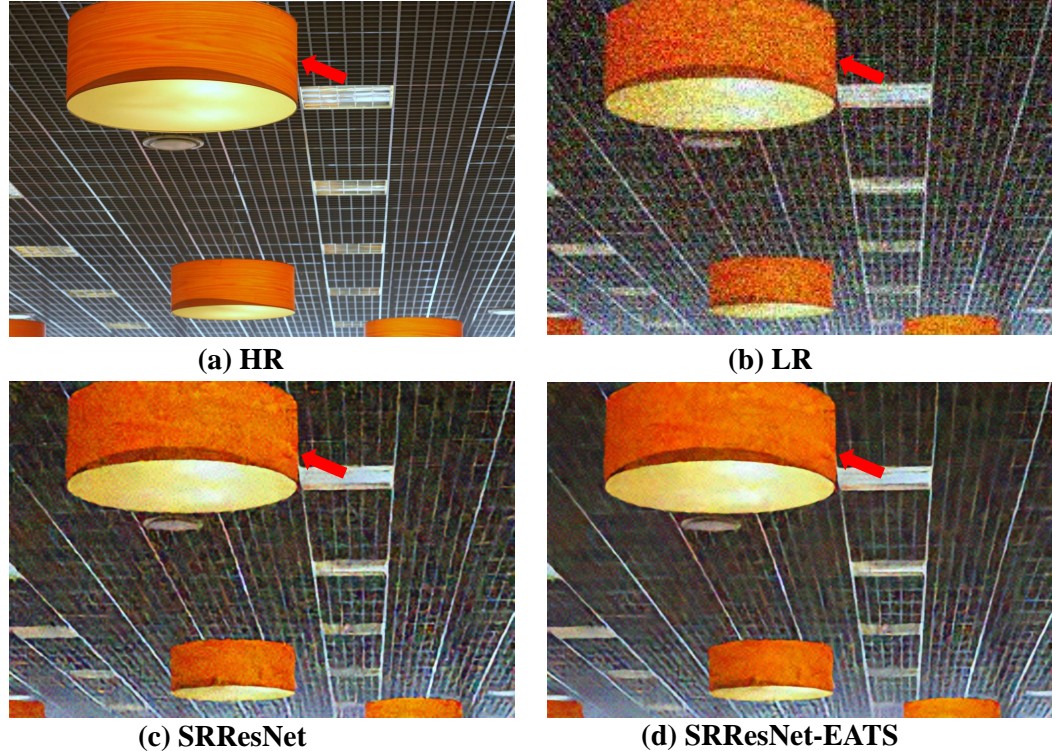

Figure 12: Visual comparison of SRResNet Ledig et al. (2017) on the noise-Urban100 Huang et al. (2015).

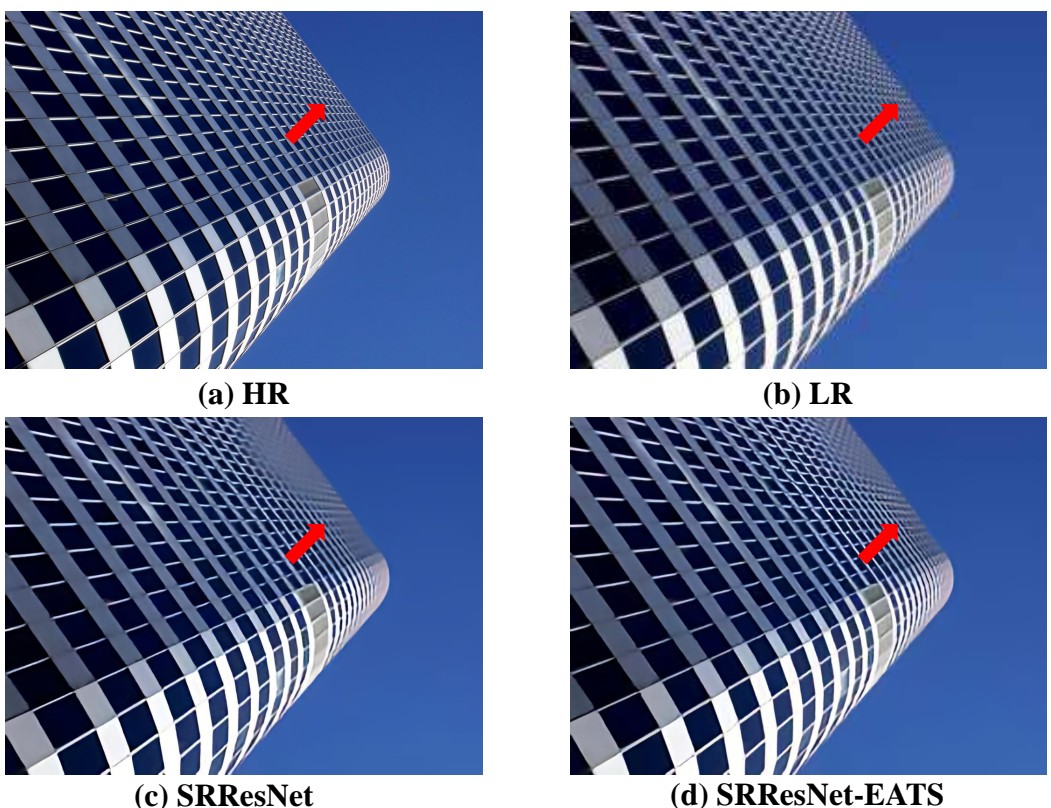

(a) HR

(b) LR

(c) SRResNet

(d) SRResNet-EATS

Figure 13: Visual comparison of SRResNet Ledig et al. (2017) on the jpeg-Urban100 Huang et al. (2015).

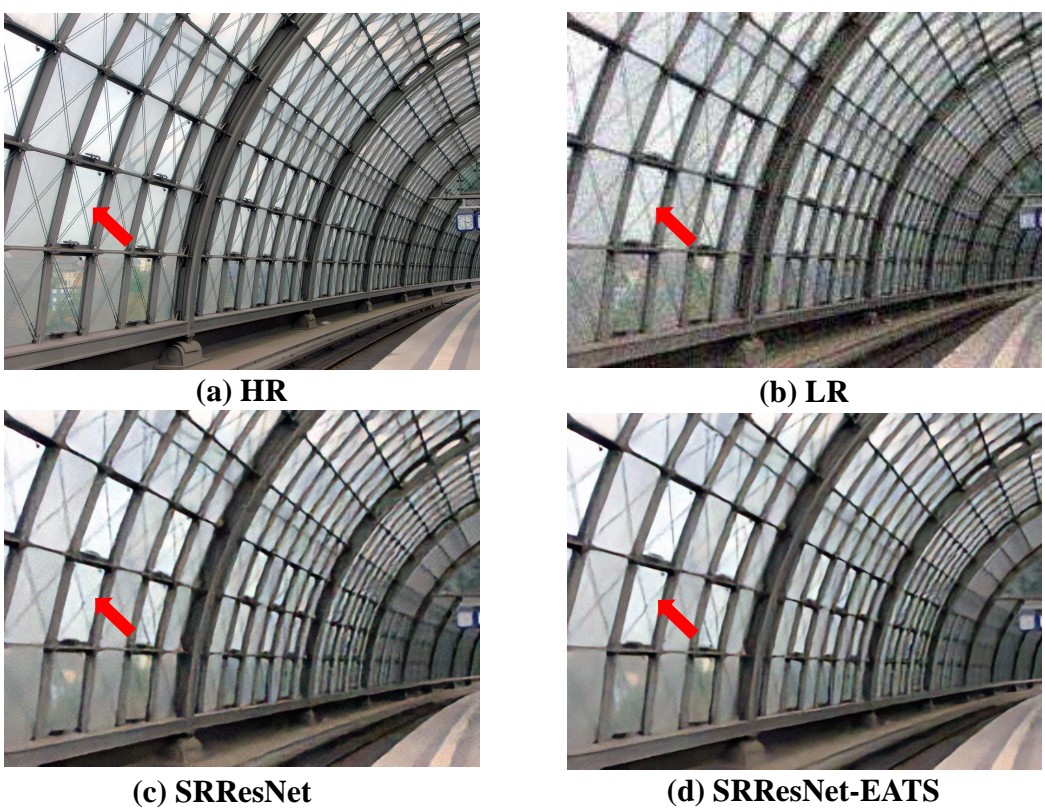

(a) HR          (b) LR

(c) SRResNet          (d) SRResNet-EATS

Figure 14: Visual comparison of SRResNet Ledig et al. (2017) on the blur-noise-Urban100 Huang et al. (2015).

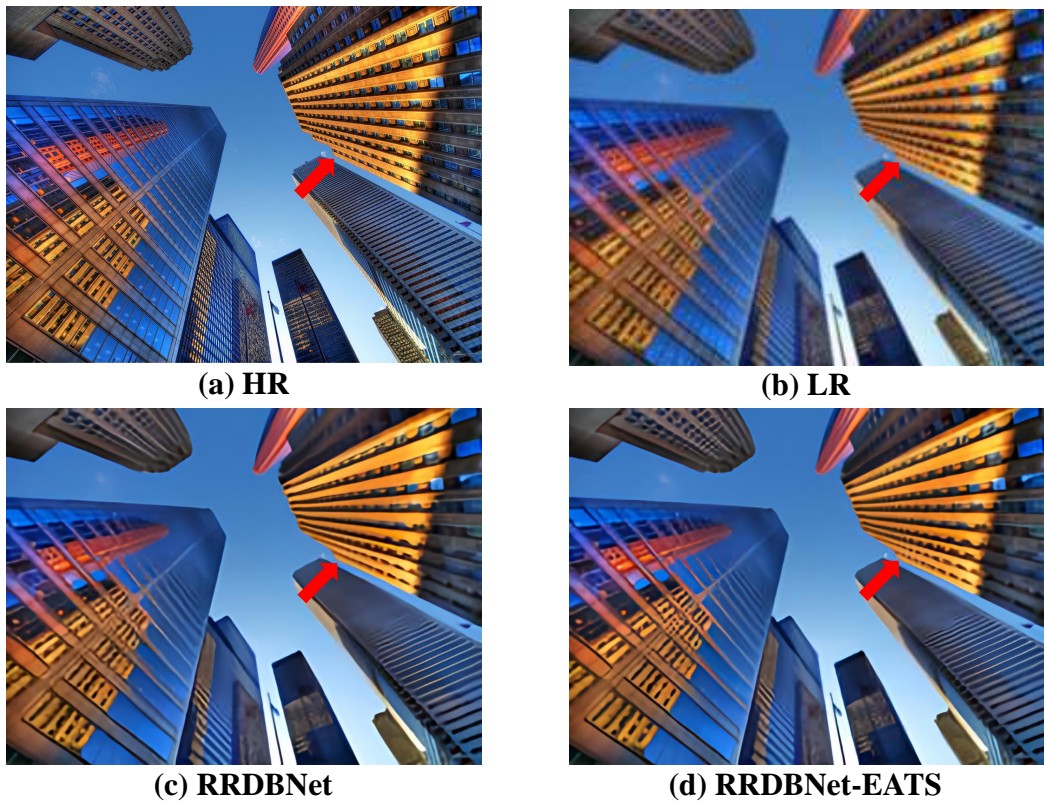

(a) HR

(b) LR

(c) RRDBNet

(d) RRDBNet-EATS

Figure 15: Visual comparison of RRDBNet Zhu et al. (2020) on the blur-jpeg-Urban100 Huang et al. (2015).

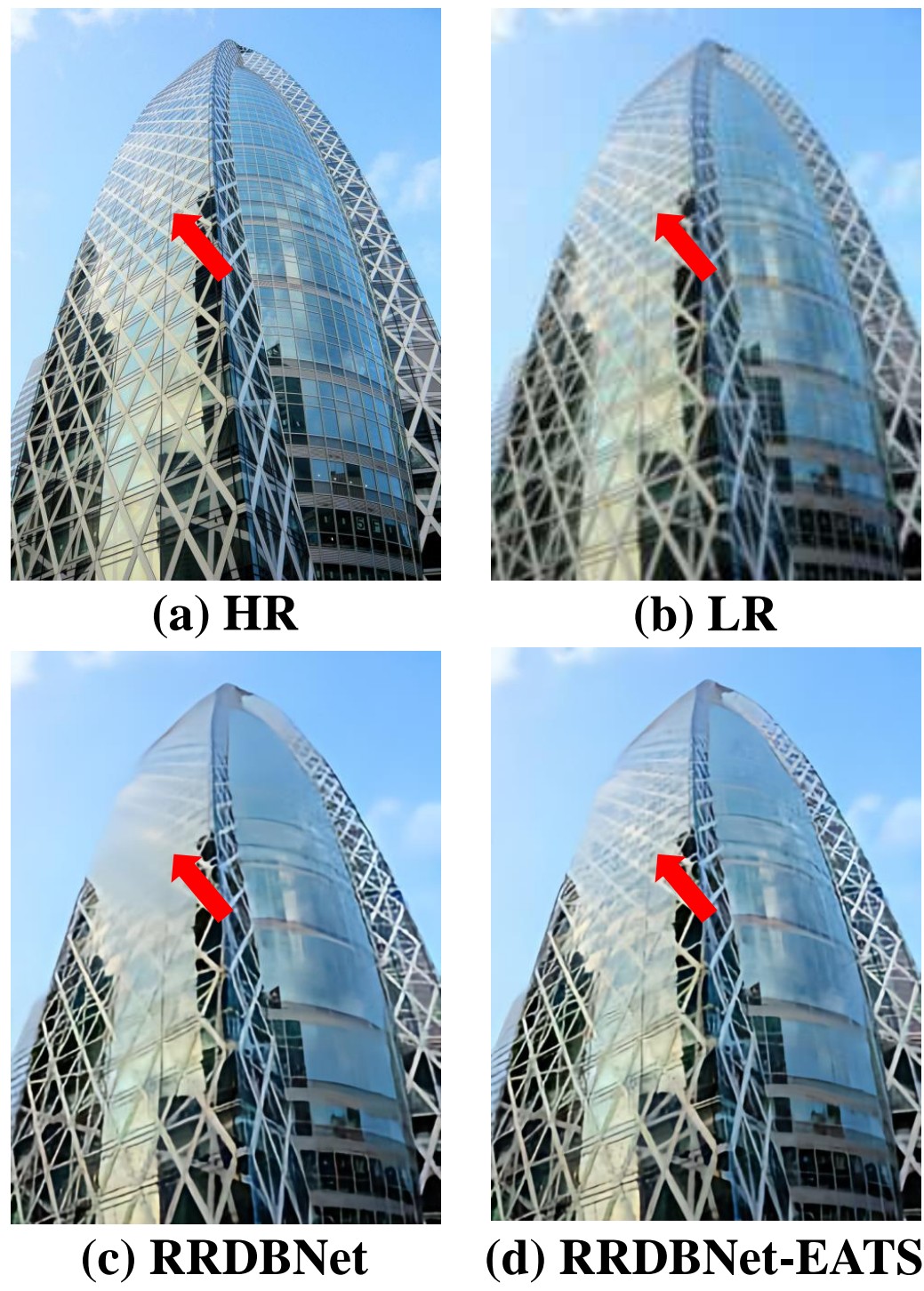

(a) HR                        (b) LR

(c) RRDBNet              (d) RRDBNet-EATS

Figure 16: Visual comparison of RRDBNet Zhu et al. (2020) on the noise-jpeg-Urban100 Huang et al. (2015).

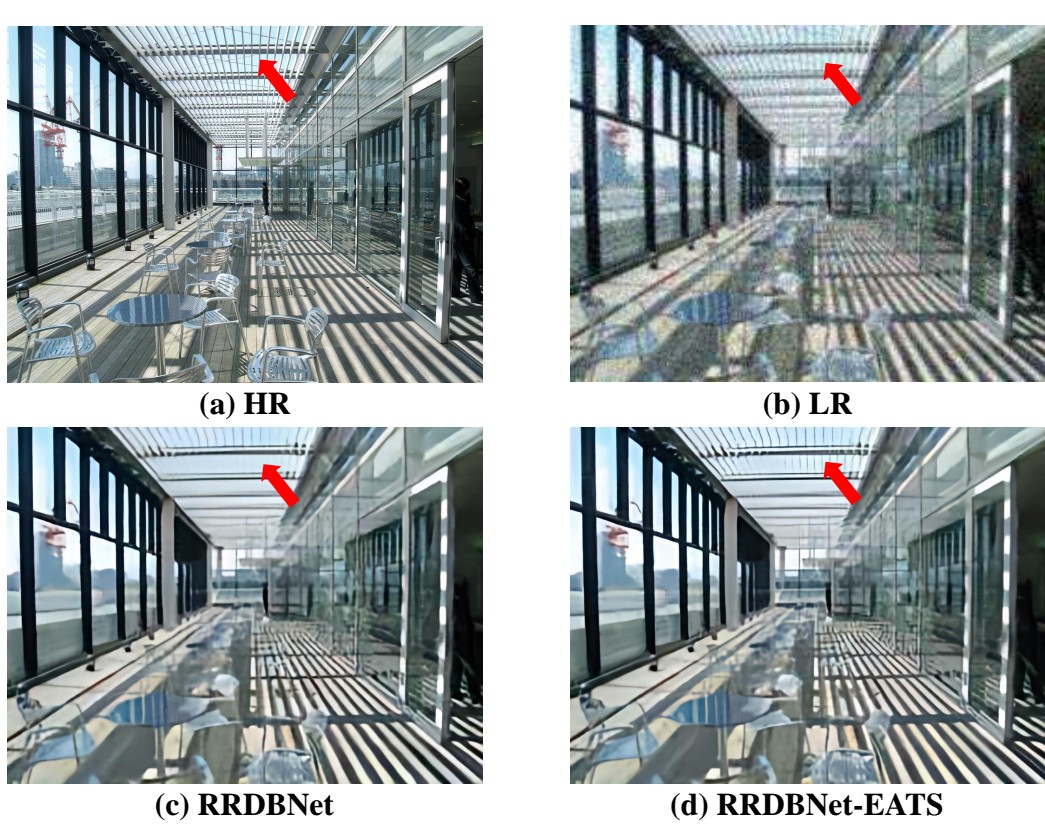

Figure 17: Visual comparison of RRDBNet Zhu et al. (2020) on the blur-noise-jpeg-Urban100 Huang et al. (2015).

