# OpenReview forum: "Awakening Collective Wisdom: Elevating Super-Resolution Network Generalization through Cooperative Game Theory"
_ICLR.cc/2024/Conference — Submitted to ICLR 2024_

### Official Review · Reviewer_inny · 2023-10-29

**Soundness:** 3 good
**Presentation:** 4 excellent
**Contribution:** 4 excellent
**Rating:** 8
**Confidence:** 4

**Summary:**

This paper attempts to awaken the suppressed filters that hinder the generalization performance for improving the generalization ability of super-resolution algorithms. To this end, the authors propose an erase-and-awaken training strategy grounded on the cooperative game theory to equitable contributions among all neurons for predictions.

**Strengths:**

1. The authors propose a novel perspective of utilizing the cooperative game theory to improve the generalization of super-resolution algorithms.
2. The proposed erase-and-awaken training strategy is general and feasible to existing SR approaches due to its simple but effective implementation through a regularization term.
3. The authors present theoretical evidence for the effectiveness of EATS in prompting equitable contributions from all neurons.
4. Through diverse analyses, including feature visualization, channel salience maps, and channel correlations, this paper effectively demonstrates the effectiveness of the proposed strategy. The comprehensive experiments and analyses support the paper’s claims and facilitate understanding of the method.

**Weaknesses:**

1. In Figure 1, the authors claim the proposed strategy alleviates the co-adaption phenomenon and achieve consistent channel response distribution. Can the dropout operation achieve the similar effect as the proposed strategy?
2. The authors have performed the ablation studies about the ratio of layers involved in EATS, indicating that the increased involvement of layers leads to better generalization. How does the number of filters involved in each training iteration affect the generalization performance?

**Questions:**

Please see the Weaknesses.

---

> ### Author Response · Authors · 2023-11-17
>
> **About dropout operation**
>
> The conclusions draw from the previous work [1] indicates that employing dropout only at the last convolutional layer improves generalization performance, while even causing drop in performance at other layers.
> Therefore, we conducted a statistical analysis of channel responses in the $2^{nd}$ and last output layers of SRResNet, SRResNet with dropout, and SRResNet-EATS (all trained on the Five5K dataset) across 100 randomly sampled unseen images.
> The visualizations are detailed in **Section H** of the Appendix.
> The results indicate that while dropout operation can alleviate the co-adaptation phenomenon in the last layer, it fails to address the issue in the shallow layers.
> In contrast, incorporating our proposed training strategy effectively mitigates co-adaptation and achieves equitable channel responses in both shallow and deep layers.
>
> [1] Kong X, Liu X, Gu J, et al. Reflash dropout in image super-resolution[C]//Proceedings of the IEEE/CVF Conference on Computer Vision and Pattern Recognition. 2022: 6002-6012.
>
> **Ablation studies**
>
> Thank you for your insightful suggestion.
> Ablation studies on the number of filters involved in each iteration are detailed in **Section F** of the Appendix.
> Our experiments explored configurations with 1 (default), 3, and 5 filters engaged in each iteration.
> Compared to the default of 1 filter, involving 3 filters achieved only a slight improvement.
> This decline may be attributed to increased randomness and instability during the training process.
>
> |         Config         |   Set5   |          |          |          |          |          |          |          |
> |:----------------------:|:--------:|:--------:|:--------:|:--------:|:--------:|:--------:|:--------:|:--------:|
> |                        |   clean  |   blur   |   noise  |   jpeg   |    b+n   |    b+j   |    n+j   |   b+n+j  |
> |        Original        | 25.2688  | 25.2776  | 22.2421  | 23.9881  | 23.3838  | 23.6890  | 23.4186  | 22.9922  |
> | EATS-1filter (default) | 26.0074  | 25.7857  | 22.5926  | 24.4301  | 23.4469  | 23.8805  | 23.6479  | 23.0888  |
> |      EATS-3filter      | 26.0538  | 25.9900  | 22.5482  | 24.4395  | 23.6294  | 23.9155  | 23.5702  | 23.0095  |
> |      EATS-5filter      | 25.8929  | 25.6534  | 22.4461  | 24.2632  | 23.5130  | 23.8032  | 23.5774  | 23.0124  |
> |          Set14         |          |          |          |          |          |          |          |          |
> |        Original        | 22.9262  | 22.8528  | 20.9357  | 22.1325  | 22.0795  | 21.9733  | 21.9026  | 21.6333  |
> | EATS-1filter (default) | 23.4922  | 23.2615  | 21.2980  | 22.4989  | 22.1894  | 22.1938  | 22.1133  | 21.7674  |
> |      EATS-3filter      | 23.6219  | 23.3394  | 21.3617  | 22.5045  | 22.3048  | 22.1650  | 22.0960  | 21.7275  |
> |      EATS-5filter      | 23.4872  | 23.1491  | 21.2607  | 22.4014  | 22.0382  | 22.1274  | 22.0476  | 21.7356  |
> |         BSDS100        |          |          |          |          |          |          |          |          |
> |        Original        | 23.3654  | 23.3880  | 21.2915  | 22.7456  | 22.3154  | 22.6020  | 22.4485  | 22.2281  |
> | EATS-1filter (default) | 23.6963  | 23.6864  | 21.5848  | 22.9928  | 22.4043  | 22.7814  | 22.5743  | 22.3039  |
> |      EATS-3filter      | 23.7743  | 23.7075  | 21.7250  | 23.0244  | 22.5536  | 22.8058  | 22.5950  | 22.3234  |
> |      EATS-5filter      | 23.5557  | 23.5699  | 21.4584  | 22.9784  | 22.5033  | 22.7022  | 22.5667  | 22.3020  |
> |        Urban100        |          |          |          |          |          |          |          |          |
> |        Original        | 21.5738  | 21.4637  | 19.5775  | 20.9649  | 20.4764  | 20.5970  | 20.7884  | 20.3406  |
> | EATS-1filter (default) | 21.9979  | 21.6462  | 19.8111  | 21.2543  | 20.5088  | 20.7175  | 20.9688  | 20.4000  |
> |      EATS-3filter      | 21.9795  | 21.7051  | 19.9301  | 21.2136  | 20.6727  | 20.6532  | 20.9733  | 20.3649  |
> |      EATS-5filter      | 21.8009  | 21.5756  | 19.7940  | 21.1980  | 20.4379  | 20.6861  | 20.9068  | 20.3656  |
> |        Manga100        |          |          |          |          |          |          |          |          |
> |        Original        | 18.6101  | 18.8597  | 18.4166  | 18.5256  | 19.0068  | 18.6980  | 18.5759  | 18.6658  |
> | EATS-1filter (default) | 19.9220  | 19.0082  | 18.6622  | 18.8107  | 19.1926  | 18.9026  | 18.7213  | 18.7592  |
> |      EATS-3filter      | 19.9446  | 19.2506  | 18.8891  | 18.8846  | 19.2921  | 19.9487  | 18.7675  | 18.7990  |
> |      EATS-5filter      | 19.8026  | 18.9324  | 18.6103  | 18.8145  | 19.1652  | 18.8039  | 18.7179  | 18.7605  |

---

### Official Review · Reviewer_5MNR · 2023-10-29

**Soundness:** 3 good
**Presentation:** 4 excellent
**Contribution:** 3 good
**Rating:** 6
**Confidence:** 5

**Summary:**

This paper proposes an erase-and-awaken training strategy to improve the generalization capacity of existing super-resolution algorithms. The strategy treats the neurons within networks as cooperative players and awakens the inhibited filters that hamper the generalization performance via an awakening regularization term. The quantitative and qualitative results show the effectiveness of the proposed strategy.

**Strengths:**

1. This paper introduces a novel perspective by conceptualizing the neurons in the network as players within a cooperative game. The authors propose an Erase-and-Awaken Training Strategy (EATS), which fosters equitable contributions from all neurons to predictions for improving model’s generalization capacity.

2. The authors establish a theoretical connection between the proposed strategy and the Shapely value in cooperative game theory. This theoretical foundation highlights the efficacy of EATS in promoting equitable contributions from all neurons.

3. The paper conducts extensive experiments and analyses, demonstrating the effectiveness of EATS. These experiments not only validate the theoretical claims but also show the practical effect of the proposed strategy.

**Weaknesses:**

1. The authors are encouraged to provide the computational costs introduced by the proposed training strategy. This addition would help reviewers to assess the practical implications and feasibility of the proposed approach.

2. The ablation studies about the impact of varying layer ratios involved in EATS make me confused. I'm uncertain about how the authors divided the involved layers. Was it through random sampling, division from shallow to deep layers, or perhaps another approach? If the division was from shallow to deep, please explain why more deep layers involved results in better performance.

**Questions:**

Please see the weaknesses.

---

> ### Author Response · Authors · 2023-11-17
>
> **Computational costs incurred by our EATS**
>
> The two forward propagations in our training paradigm marginally increase training times, approximately by 1.4 times compared to the corresponding original networks.
> This increment is deemed acceptable, especially considering the significant improvement in generalization capability.
> Additionally, it's important to highlight that our training paradigm is exclusively applied during the training process and does not impose any additional computational burden during the inference phase.
> We have added the discussion in the **Section C** of Appendix.
>
> **Clarification about ablation studies about the impact of varying layer ratios involved in EATS**
>
> Thank you for bringing up this concern. The involved layers in our proposed EATS are randomly sampled in advance before the training process.
> Subsequently, the EATS is applied to the selected layers.
> The experimental results consistently demonstrate that involving more layers leads to better generalization performance. This aligns with our objective of awakening a greater number of neurons and promoting equitable contributions across all neurons.

---

### Official Review · Reviewer_1q9R · 2023-10-31

**Soundness:** 3 good
**Presentation:** 3 good
**Contribution:** 3 good
**Rating:** 6
**Confidence:** 5

**Summary:**

This paper introduces the cooperative game theory improve the generalization of super-resolution algorithms. It proposes an erase-and-awaken training strategy, which prompts all neurons in networks to achieve equitable contributions to predictions by an awakening regularization term. This paper conducts efficient analyses of feature responses to demonstrate the effectiveness of the proposed EATS.

**Strengths:**

1. This paper improve the generalization capability of image super-resolution algorithms from a novel perspective of the cooperative game theory. It views all neurons within the network as the active participants in a cooperative game.

2. The authors propose an Erase-and-Awaken Training Strategy (EATS) to awaken the inhibited neurons that hinder the generalization performance. EATS promotes equitable contributions from all neurons to the predictions, thus improving the generalization capability of networks on the unseen scenarios.

3. The authors provide the theoretical proof to validate the effectiveness of EATS in improving the Shapely value, which signifies the contribution of each participant to predictions. In addition, authors present efficient analyses about feature responses, providing substantial evidence to support the claims made in this paper.


=========================== Update

I have read the author feedback and the reviews from other reviewers. I keep my rating as 6--accept

**Weaknesses:**

1. The implementation of EATS involves an awakening regularization term, which emphasizes the contribution of the erased filter by constraining predictions of the disrupted network approximates the baseline image. However, the random sampling of layers and filters in each training iteration has raised concerns about the convergence of the awakening regularization term. Therefore, the authors are encouraged to provide the curve of the regularization in the training process.

2. The proposed EATS requires an additional forward propagation step during each training iteration. It is crucial to quantify the extra training time incurred by this strategy compared to the original baseline algorithm.

**Questions:**

Figure 1 in the manuscript presents the channel responses of the shallow layer, 2nd block, in SRResNet and SRResNet-EATS. However, it would be valuable for reviewers to also observe if similar phenomena occur in deep layers, where channel responses are awakened and equitable. Visualizations of channel responses from shallow to deep layers would provide a more comprehensive understanding of the awakening effect across various network depths.

---

> ### Author Response · Authors · 2023-11-17
>
> **The convergence of the awakening regularization**
>
> Thanks for the suggestion. We have incorporated the visualizations of the original loss, $L_{ori}$, and our awakening regularization, $L_{awa}$, during the training process of SRResNet-EATS and RRDBNet-EATS.
> These visualizations can be found in **Section D** of Appendix.
> Through the plots, it is evident that the awakening regularization progressively decreases and converges with the training iteration.
>
> **Training time incurred by our EATS**
>
> The two forward propagations in our training paradigm marginally increase training times, approximately by 1.4 times compared to the corresponding original networks.
> This increment is deemed acceptable, especially considering the significant improvement in generalization capability.
> Additionally, it's important to highlight that our training paradigm is exclusively applied during the training process and does not impose any additional computational burden during the inference phase.
> We have added the discussion in the **Section C** of Appendix.
>
> **Activating channel responses in the deep layer**
>
> We conducted an analysis of channel responses from a deep layer, specifically the output block, in SRResNet (trained on Five5K) on 100 randomly sampled unseen images.
> The visualizations are presented in **Section H** of the Appendix.
> Similar to the observed phenomenon in the shallow layer, the deep layer of the SRResNet exhibits co-adaptation, where a few channels are highly activated while others are inhibited.
> In contrast, SRResNet incorporated with our EATS demonstrates equitable and activated channel responses.
> This indicates that our proposed training strategy is effective in alleviating the co-adaptation and activating channel responses from shallow to deep layers.

---

### Official Review · Reviewer_P3P8 · 2023-11-01

**Soundness:** 3 good
**Presentation:** 3 good
**Contribution:** 3 good
**Rating:** 6
**Confidence:** 3

**Summary:**

The following paper proposes Erase-and-Awaken Training Strategy (EATS), a cooperative game theory-inspired novel training strategy that improves the generalization capability for Single Image Super-Resolution (SISR) algorithms in dealing with real-world scenarios. Unlike previous data-driven methods, EATS presents itself as an optimization-based method, where it encourage all neurons within an existing SISR framework to actively collaborate in solving the generalization problem by randomly perturbing response of inhibited neurons and maximize their contributions to prediction. EATS consists of two steps: (1) **Erasing step**, where it randomly samples a disruptor filter $n_l^{dis}$ and applies it towards a randomly selected erased filter $n_l^{ear}$ in the network's $l^{th}$ layer and assess performance before-and-after the erasure ($f_{\theta}$ and $f_{\theta'}$), and (2) **Awakening step**, where an awakening regularization term $\mathcal{L}\_{awa}$ is employed to close the gap between predicted high-resolution image and the low-resolution image from the disrupted network $f_{\theta'}$. Besides providing theoretical proof to show the effectiveness of EATS in improving the Shapley value of the network, experiments on multi-degradation settings within various unseen datasets have demonstrated that plugging EATS into SISR algorithms such as SRResNet and RRDBNet results in quantitatively and qualitatively better images than baselines.

**Strengths:**

- The proposed method is original, simple, and applicable to existing methods.
- The paper is well-written for the most part.
- Incorporating EATS with existing baseline methods SRResNet [1] and RRDBNet [2] outperforms the baselines on various tasks and baseline + Dropout for the most part.

**Weaknesses:**

Aside from the limitations pointed out in Section 5 of the paper, I have several concerns regarding the paper:

- The experiments could have used more recent SISR models instead of SRResNet and RRDBNet, which have been more than 5 years old. One possible way to alleviate this issue is to replicate Table 1 and Table 2 results in the paper using more recent attention-based architectures like HAN [3], SAN [4], or SwinIR [5] to demonstrate the effectiveness of EATS, if possible.
- Minor typos exist in the text. For instance, 'Shapely' should be 'Shapley' in the abstract and the Solution subpoint on the 2nd page. In addition to that, 'Managa109' should be 'Manga109' [6] on Table 2 on the 9th page.

**Questions:**

To my understanding, since the model uses two networks: original network $f_{\theta}$ and disrupted network $f_{\theta'}$, does that mean that the whole framework involves more parameters (approximately doubles) to that of the baselines? If yes, can authors address the # of parameters involved in employing said method with EATS?

[1] Ledig, Christian, et al. "Photo-realistic single image super-resolution using a generative adversarial network." Proceedings of the IEEE conference on computer vision and pattern recognition. 2017. https://arxiv.org/abs/1609.04802

[2] Wang, Xintao, et al. "ESRGAN: Enhanced Super-Resolution Generative Adversarial Networks." Proceedings of the European conference on computer vision (ECCV) workshops. 2018. https://arxiv.org/abs/1809.00219

[3] Niu, Ben, et al. "Single image super-resolution via a holistic attention network." Computer Vision–ECCV 2020: 16th European Conference, Glasgow, UK, August 23–28, 2020, Proceedings, Part XII 16. Springer International Publishing, 2020. https://arxiv.org/abs/2008.08767

[4] Dai, Tao, et al. "Second-order attention network for single image super-resolution." Proceedings of the IEEE/CVF conference on computer vision and pattern recognition. 2019. https://ieeexplore.ieee.org/document/8954252

[5] Liang, Jingyun, et al. "SwinIR: Image Restoration Using Swin Transformer." Proceedings of the IEEE/CVF international conference on computer vision. 2021. https://arxiv.org/abs/2108.10257

[6] Matsui, Yusuke, et al. "Sketch-based manga retrieval using manga109 dataset." Multimedia Tools and Applications 76 (2017): 21811-21838. https://arxiv.org/abs/1510.04389

---

> ### Author Response · Authors · 2023-11-17
>
> **Evaluation on attention-based algorithm**
>
> Thank you for your valuable suggestion.
> Given the time constraints, we have evaluated the effectiveness of our proposed EATS using a lightweight variant of HAN [1] with x4 scaling.
> This lightweight version comprises approximately 1/20 of the parameters found in the original network.
> To maintain consistency with the experimental settings outlined in the manuscript, we have re-trained both the lightweight HAN and HAN-EATS, incorporating high-order degradation modeling.
> The quantitative results are presented below. Additionally, we plan to conduct further experiments on the x2 scale and will include the updated results in the **Section B** of Appendix upon acceptance of this paper.
>
> |    Config   |   Set5   |          |          |          |          |          |          |          |
> |:-----------:|:--------:|:--------:|:--------:|:--------:|:--------:|:--------:|:--------:|:--------:|
> |             |   clean  |   blur   |   noise  |   jpeg   |    b+n   |    b+j   |    n+j   |   b+n+j  |
> |   Original  | 25.4834  | 25.4207  | 22.5964  | 24.1479  | 23.7007  | 23.8615  | 23.5668  | 23.1275  |
> |     EATS    | 26.1176  | 25.8898  | 22.8330  | 24.4354  | 23.8923  | 23.9615  | 23.6703  | 23.1699  |
> | Improvement |  0.6342  |  0.4691  |  0.2366  |  0.2875  |  0.1916  |  0.1000  |  0.1035  |  0.0424  |
> |    Set14    |          |          |          |          |          |          |          |          |
> |   Original  | 23.0610  | 23.1144  | 21.3902  | 22.2880  | 22.3199  | 22.2746  | 22.0212  | 21.7860  |
> |     EATS    | 23.4764  | 23.3560  | 21.5058  | 22.4861  | 22.3971  | 22.2772  | 22.1089  | 21.7872  |
> | Improvement |  0.4154  |  0.2416  |  0.1156  |  0.1981  |  0.0772  |  0.0026  |  0.0877  |  0.0012  |
> |   BSDS100   |          |          |          |          |          |          |          |          |
> |   Original  | 23.4619  | 23.4352  | 21.7799  | 22.8227  | 22.5801  | 22.7466  | 22.4817  | 22.2781  |
> |     EATS    | 23.6903  | 23.5756  | 21.8599  | 22.9232  | 22.6394  | 22.7573  | 22.5339  | 22.2937  |
> | Improvement |  0.2284  |  0.1404  |  0.0800  |  0.1005  |  0.0593  |  0.0107  |  0.0522  |  0.0156  |
> |   Urban100  |          |          |          |          |          |          |          |          |
> |   Original  | 21.6242  | 21.5168  | 20.2169  | 21.0329  | 20.7608  | 20.7810  | 20.7895  | 20.3836  |
> |     EATS    | 21.9152  | 21.7400  | 20.2708  | 21.1839  | 20.7677  | 20.8224  | 20.8381  | 20.3987  |
> | Improvement |  0.2910  |  0.2232  |  0.0539  |  0.1510  |  0.0069  |  0.0414  |  0.0486  |  0.0151  |
> |   Manga100  |          |          |          |          |          |          |          |          |
> |   Original  | 18.9412  | 19.2265  | 18.6995  | 18.7634  | 19.1281  | 19.0087  | 18.7372  | 18.7624  |
> |     EATS    | 19.3123  | 19.5149  | 18.8154  | 19.0205  | 19.2600  | 19.1412  | 18.8486  | 18.9039  |
> | Improvement |  0.3711  |  0.2884  |  0.1159  |  0.2571  |  0.1319  |  0.1325  |  0.1114  |  0.1415  |
>
> [1] Niu, Ben, et al. "Single image super-resolution via a holistic attention network." Computer Vision–ECCV 2020: 16th European Conference, Glasgow, UK, August 23–28, 2020, Proceedings, Part XII 16. Springer International Publishing, 2020. https://arxiv.org/abs/2008.08767
>
> **Typos**
>
> Thank you for bringing these typos to our attention. We appreciate your keen observation. The corrections have been addressed in the revision.
>
> **Clarification regarding the parameters involved in our training paradigm**
>
> Firstly, it's essential to emphasize that our proposed Erasing and Awakening Training Scheme (EATS) does not introduce additional training parameters.
> Within a training iteration, we conduct a forward propagation of the original network, $f_θ$, calculating the original loss function, $L_{ori}$.
> Subsequently, our EATS involves randomly replacing an erased filter with a disrupted filter based on the original network, resulting in a modified network after erasure, $f_{θ^{'}}$.
> We then calculate the awakening regularization, $L_{awa}$, through a second forward propagation.
> The network is update based on the combination of these two loss functions.
> Therefore, it's crucial to note that our EATS only requires two forward propagations, as opposed to employing two separate neural networks.
>
> Secondly, the two forward propagations in our training paradigm marginally increase training times, approximately by 1.4 times compared to the corresponding original networks.
> This increment is deemed acceptable, especially considering the significant improvement in generalization capability.
>
> Lastly, it's important to highlight that our training paradigm is exclusively applied during the training process and does not impose any additional computational burden during the inference phase.
>
> We have added the discussion in the **Section C** of Appendix.

---

> > ### Comment · Reviewer_P3P8 · 2023-11-19
> >
> > Dear Authors,
> >
> > Thank you for responding with additional results on attention-based SISR algorithms and clarifications regarding the number of parameters within EATS.
> >
> > The results are convincing for attention-based algorithms in HAN, showing how EATS applies to a wider range of network architectures. Furthermore, we can observe that EATS imposes an acceptable addition of computational overhead in comparison to the existing SISR algorithm.
> >
> > With that in mind, I would like to increase my score.

---

### Meta-Review · Program_Chairs · 2023-12-05

**Metareview:**

This review is provided by the PCs.  After calibration and downweighted inflated and non-informative reviews, this paper was deemed below the accept threshold.

This paper proposes an erase and awaken training strategy (EATS) to improve super-resolution networks.  The algorithm essentially swaps two neurons at random in a randomly chosen layer, leading to a perturbed network .  The loss function is defined as improvement in the perturbed network's ability to reconstruct a bicubic upsampling of the low-res image (why not the high-res image?), and the original network's ability to reconstruct the high-res image.  The paper then draws an analogy to the Shapley value in cooperative game theory, and evaluate on a range of settings.

The PCs found the paper to be poorly justified.  For example, theory section (Section 3.3) is not actually a formal proof.  There is not statement of a theorem.  Moreover, that section never defines what the utility function v is (e.g., is it the reconstruction loss w.r.t. the bicubic interpolation?).  Why does one even need to draw a connection to Shapley and cooperative game theory in the first place, when it's much more natural to view this approach as a variant of drop-out type approaches that perturb the network (and thus have some kind of regularization effect)?  In addition, the algorithm itself has a peculiar design choice of using the bicubic upsampling to induce a gradient, and this design choice was never discussed.

The empirical results show very modest performance gains, which by itself is not enough to justify accept.

**Justification For Why Not Higher Score:**

Please refer to the metareview.

**Justification For Why Not Lower Score:**

Please refer to the metareview.

---

### Decision · Program_Chairs · 2024-01-16

Reject